METHODS AND RESOURCES

# Sensitive red fluorescent indicators for real-time visualization of potassium ion dynamics in vivo

**Lina Yang**[1,2,3,4], **Vishaka Pathiranage**[5☉], **Shihao Zhou**[2,3,4☉], **Xiaoting Sun**[2,3,4☉],
**Hanbin Zhang**[2,3,4], **Cuixin Lai**[2,3,4], **Chenlei Gu**[2,3,4], **Fedor V. Subach**[6], **Mikhail Drobizhev**[7],
**Alice R. Walker**[5]*, **Kiryl D. Piatkevich** [2,3,4]*

1 School of Life Sciences, Fudan University, Shanghai, China, 2 School of Life Sciences, Westlake University, Hangzhou, Zhejiang, China, 3 Westlake Laboratory of Life Sciences and Biomedicine, Hangzhou, Zhejiang, China, 4 Institute of Basic Medical Sciences, Westlake Institute for Advance Study, Hangzhou, Zhejiang, China, 5 Department of Chemistry, Wayne State University, Detroit, Michigan, United States of America, 6 Complex of NBICS Technologies, National Research Center "Kurchatov Institute", Moscow, Russia, 7 Department of Microbiology and Cell Biology, Montana State University, Bozeman, Montana, United States of America

☉ These authors contributed equally to this work.
* arwalker@wayne.edu (ARW); kiryl.piatkevich@westlake.edu.cn (KDP)

## Abstract

Potassium ion (K⁺) dynamics are vital for various biological processes. However, the limited availability of detection tools for tracking intracellular and extracellular K⁺ has impeded a comprehensive understanding of the physiological roles of K⁺ in intact biological systems. In this study, we developed two novel red genetically encoded potassium indicators (RGEPOs), RGEPO1 and RGEPO2, through a combination of directed evolution in *Escherichia coli* and subsequent optimization in mammalian cells. RGEPO1, targeted to the extracellular membrane, and RGEPO2, localized in the cytoplasm, exhibited positive K⁺-specific fluorescence response with affinities of 2.4 and 43.3 mM in HEK293FT cells, respectively. We employed RGEPOs for real-time monitoring of subsecond K⁺ dynamics in cultured neurons, astrocytes, acute brain slices, and the awake mouse in both intracellular and extracellular environments. Using RGEPOs, we were able, for the first time, to visualize intracellular and extracellular potassium transients during seizures in the brains of awake mice. Furthermore, molecular dynamics simulations provided new insights into the potassium-binding mechanisms of RGEPO1 and RGEPO2, revealing distinct K⁺-binding pockets and structural features. Thus, RGEPOs represent a significant advancement in potassium imaging, providing enhanced tools for real-time visualization of K⁺ dynamics in various cell types and cellular environments.

## Introduction

Potassium ion (K⁺) homeostasis and dynamics play critical roles in a multitude of biological activities in all domains of life [1,2]. For example, in neurons, K⁺ is vital for

**Data availability statement:** All essential raw datasets, including source files for the S1 Data, supplementary figures, and raw unprocessed ND2 images of cultured neurons, have been deposited in Figshare (https://doi.org/10.6084/m9.figshare.29917037). All plasmids used in this study are available from WeKwikGene (https://wekwikgene.wllsb.edu.cn/publications/0000741; barcodes: 0000741, 0000742, 0000765, 0000766, 0001230, 0001231). The computational inputs for molecular dynamics simulations, including topology files, initial coordinates of all systems, the last frames of the free diffusion simulations of both RGEPO1 and RGEPO2, force field parameters for the chromophore, and a short simulation video of potassium diffusion in RGEPO1, have been deposited in Zenodo (https://doi.org/10.5281/zenodo.13824581).

**Funding:** The work was supported by start-up funding from the Foundation of Westlake University, Westlake Laboratory of Life Sciences and Biomedicine, National Natural Science Foundation of China grant 32171093, and "Pioneer" and "Leading Goose" R&D Program of Zhejiang 2024SSYS0031 to K.D.P. The authors thankfully acknowledge support from the National Science Foundation–NSF through the grant NSF CHE2338804 to A.R.W. We thank Wayne State University and Department of Chemistry for the Thomas C. Rumble University Graduate Fellowship to V. P. M.D. thanks the NIH/NINDS BRAIN Initiative grant 2U24NS107109 for supporting his Resource for Multiphoton Characterization of Genetically Encoded Probes. F.V.S. acknowledges that the work was carried out within the state assignment of NRC "Kurchatov Institute" (design part of the sensor). The funders had no role in study design, data collection and analysis, decision to publish, or preparation of the manuscript.

**Competing interests:** The authors have declared that no competing interests exist.

**Abbreviations:** DIV, days in vitro; DMEM, Dulbecco's modified Eagle's medium; EC, effective extinction coefficients; FBS, fetal bovine serum; FP, fluorescent protein; FRET, Förster Resonance Energy Transfer; GEPOs, genetically encoded potassium indicators; K$^+$, potassium ion; MD, molecular dynamics; NCBI, National Center for Biotechnology Information; PCR, polymerase chain reaction; PDGFRβ, platelet-derived growth factor receptor; QYs, quantum yields; RGEPOs, red genetically encoded potassium indicators; ROI, region of interest.

generating action potentials and maintaining the resting membrane potential, which is critical for proper signal transmission [3,4]. Astrocytes, on the other hand, regulate extracellular K$^+$ levels through a process known as potassium buffering, which is essential for preventing neuronal hyperexcitability and maintaining overall ion homeostasis [5]. Under resting conditions in the mammalian cells, K$^+$ has an intracellular concentration of 142–175 mM, while extracellular concentration is in the range of 3.5–5 mM, maintaining the resting membrane potential at −70 to −80 mV [6,7]. Therefore, sensitive ion probes with a high dynamic range at physiological concentrations are essential for accurately analyzing K$^+$ homeostasis and its functional transients.

Traditionally, K$^+$ levels have been measured using ion-sensitive electrodes [8,9] or fluorescent dyes [10–12]. While ion-sensitive electrodes provide direct and precise measurements of absolute K$^+$ concentrations [9], they are highly invasive and typically offer limited spatial resolution and throughput of measurements, making them unsuitable for studying fine-scale cellular processes in intact cells and tissues. Fluorescent dyes, such as potassium-sensitive fluorophores, offer improved spatial resolution compared to electrodes and allow for noninvasive imaging of K$^+$ levels in live cells [10,12]. However, they suffer from drawbacks such as poor ion selectivity, labeling specificity, limited dynamic range, cytotoxicity, and lack of targetability to specific cell types or subcellular compartments.

Genetically encoded potassium indicators (GEPOs) provide a promising alternative for K$^+$ measurement, allowing for real-time, minimally invasive monitoring with high spatial and temporal resolution [13–16]. Recent achievements in developing GEPOs stemmed from using a small water-soluble potassium-binding protein from *Escherichia coli*, known as Ec-Kbp (formerly YgaU) [17]. Among these indicators, GEPII [16] and KIRIN1s [15] are based on the Förster Resonance Energy Transfer (FRET) (S1 Table). The FRET-based GEPOs provide ratiometric output, which enhances the accuracy of concentration measurements. However, they require two specific emission color bands, limiting their application in multiplexed imaging experiments. Additionally, FRET-based indicators often suffer from weak signals and limited dynamic range. In contrast, single fluorescent protein (FP)-based indicators, although more challenging to engineer than FRET-based biosensors and typically intensiometric, require only single-color bands for excitation and emission. This makes them well-suited for multiparameter imaging [18], and they are generally characterized by higher dynamic range and stronger signals. The green GEPOs, GINKO1 [15], GINKO2 [13], and KRaIONs [14], based on the single GFP-like FPs, have been successfully applied to monitor intracellular K$^+$ dynamics in mammalian cells and plants (S1 Table). Compared to green fluorescence proteins, red fluorescence proteins provide the advantage of deeper tissue penetration, low phototoxicity, and compatibility with a large variety of existing green indicators. Thus, the development of red fluorescent GEPOs (RGEPOs) is essential for advancing K$^+$ research.

By integrating insights garnered from genomic mining and informed by structural guidance in potassium-binding protein architecture, we successfully developed two red fluorescent potassium indicators, designated as RGEPO1 and RGEPO2. These indicators were engineered by inserting a potassium-binding protein cloned from a

PLOS Biology

hydrothermal vent bacterium into the FP mApple at β-strand 7 next to the chromophore. RGEPO1 and RGEPO2 exhibited a ΔF/F in the range of 400%–1,000% in solution and 100%–300% in HEK293FT cells in response to 150 and 200 mM K$^+$, respectively. In HEK293FT cells, potassium affinities were determined to be 3.55 mM for RGEPO1 and 14.81 mM for RGEPO2. Additionally, we discovered the molecular mechanisms underlying the different responses of RGEPO1 and RGEPO2 to potassium ions using molecular dynamic simulation. Furthermore, RGEPOs were applied to monitor intracellular and extracellular K$^+$ dynamics in the culture neurons, astrocytes, acute brain slices, and mouse seizures in vivo. These findings substantiate the practical utility of indicators RGEPO1 and RGEPO2 as essential tools for the real-time visualization and comprehensive analysis of intracellular and extracellular potassium ion dynamics.

## Results

### Development and validation of red fluorescent potassium indicator

To engineer a single red FP-based K$^+$ indicator, we initiated our efforts by identifying a large diversity of putative naturally occurring potassium-binding proteins, aiming to develop a sensor with a larger dynamic range that can be effectively applied across various biological contexts. For genome mining, we used protein BLAST on the 'env_nr' dataset. This dataset includes proteins from whole genome shotgun sequencing (WGS) metagenomic projects, which was searched using the Hv-Kbp protein (see the original publication reporting its function [14]) as a query sequence. The search identified five proteins with putative functions that shared 40%−70% amino acid identity with the Hv-Kbp protein and had conserved K$^+$ binding sites based on the primary sequence alignment with the Ec-Kbp protein, which has confirmed potassium binding pocket with 6 amino acids coordinating the cation (S1 Fig). To validate their potassium binding properties, we swapped Ec-Kbp in the KRaION1 indicator [14] with each of these homologs and expressed them in *E. coli* to assess sensitivity to K$^+$ (Fig 1A). All generated constructs exhibited positive fluorescence responses to K$^+$, confirming their function as K$^+$-binding proteins (S2 Fig). Among them, Hv-Kbp-mNG and MNT-mNG displayed the largest ΔF/F values, reaching 59%, and 62%, respectively, while also exhibiting the highest baseline fluorescence (Fig 1B) making them primary candidates, along with Ec-Kbp, for developing red potassium indicators.

To construct a red fluorescence potassium sensor, we inserted these two homologs as well as Ec-Kbp into the mApple FP between amino acids S151 and V152 derived from the previously developed and characterized FRCaMP calcium sensor [19–21] (following the previous design of red Ca$^{2+}$ sensor FRCaMP [19] but with inverted topology as described in ref. [20]; Fig 1C) and expressed the constructs in *E. coli* for further characterization. All three variants were functional, although they exhibited limited ΔF/F below 30% with comparable baseline brightness (Fig 1B). Considering the brightness and ΔF/F, we selected the Hv-Kbp-based chimera for further optimization through directed molecular evolution in *E. coli*, which allows for rapid and efficient high-throughput screening of libraries to enhance the prototype [13,22] (Fig 1D). Previous studies on high-performance biosensors revealed the critical role of linkers between sensing and reporting moieties in biosensor functionality and performance [23,24]. Therefore, we started by optimizing the length and amino acid composition of the linkers near the insertion site, ultimately identifying a prototype sensor with linker sequences Linker1: P and Linker2: SAP (S3A Fig). This prototype exhibited a robust fluorescence increase with ΔF/F of 418% in response to 230 mM K$^+$ (Fig 1E, Round 2). Further optimization using random mutagenesis, targeting both the binding domain and the entire sensor sequence, produced variants with maximum ΔF/F values exceeding 600% (Figs 1E and S3B).

Sensor properties in solution are not always readily translated to mammalian cells [22]. Since our goal was to develop a sensor suitable for imaging mammalian cells, we selected the top-performing variants identified through *E. coli* screening for side-by-side comparison in HEK293FT cells. To evaluate these candidates in a cellular environment, we expressed and assessed them in HEK293FT cells, titrating with 200 mM K$^+$ in the presence of 15 μM valinomycin and 5 μM CCCP, which facilitated K$^+$ influx and maintained a stable pH (Fig 2A). This K$^+$ concentration, slightly above typical levels in mammalian cells [25], was chosen to enhance the interaction between potassium ions and the indicator, ensuring a robust and detectable response for effective screening. Among these, RGEPO0.5 produced a peak change in fluorescence of

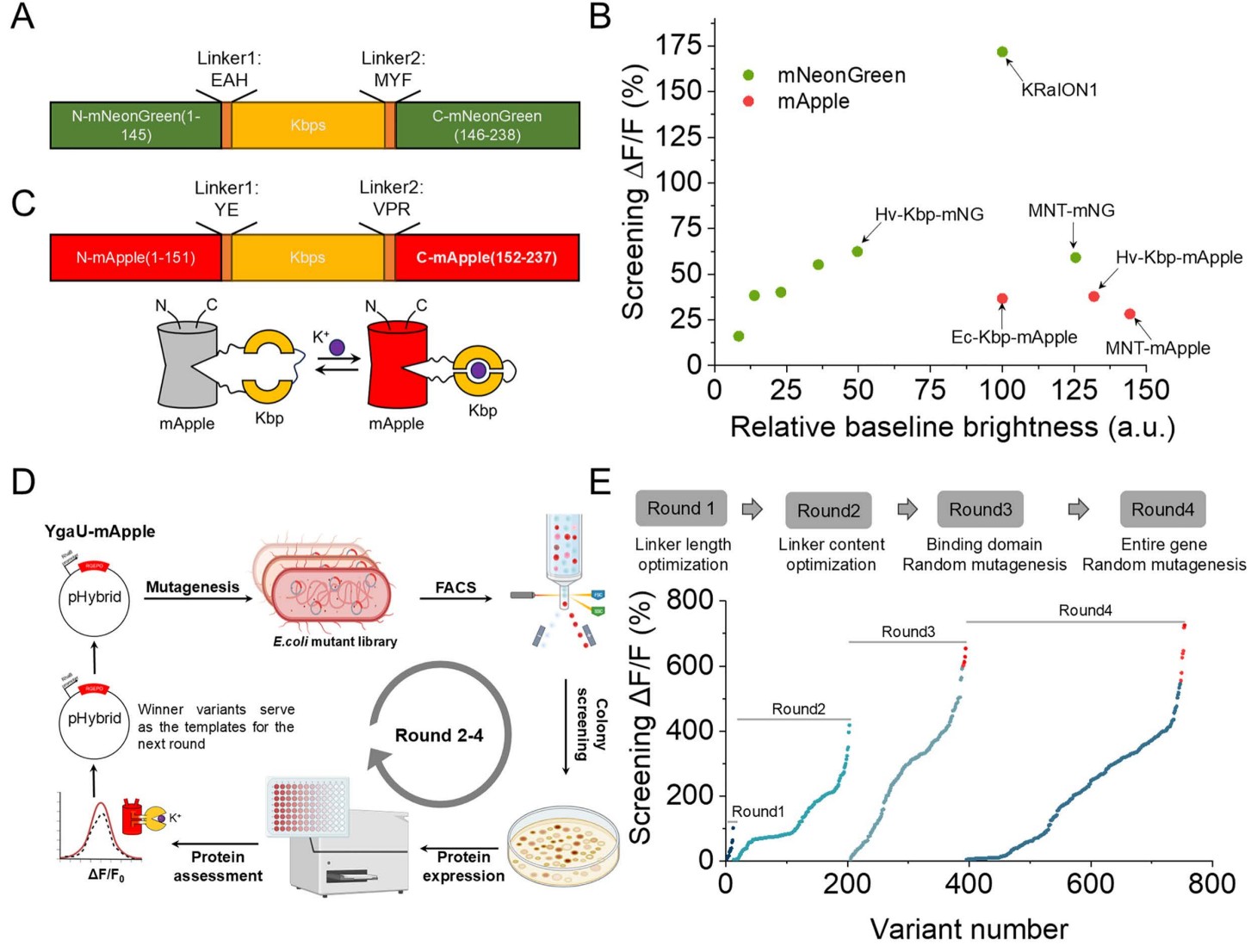

**Fig 1. Development and optimization of red potassium biosensors in *E. coli*. (A)** Schematic representation of framework of Kbp's homologes inserted into mNeoGreen. **(B)** Screening Δ*F/F* vs. screening brightness rank plot representing the homologes of Kbp tested in *Escherichia coli* based on mNeoGreen and mApple, respectively. **(C)** Schematic representation of RGEPO and its putative mechanism of response to K⁺. **(D)** Schematic depiction of the random mutagenesis workflow. Random mutagenesis was applied to the binding domain or the entire gene of the template, yielding a mutant library. Generated mutants were selected using FACS based on brightness. Protein extraction was carried out, and ΔF/F was determined upon the addition of 200 mM K⁺. The variants exhibiting the highest ΔF/F were employed as the template for the next round. FACS, fluorescence-activating cell sorting (created in BioRender.com). **(E)** In vitro stages in the identification of potential RGEPO candidates upon the addition of 230 mM K⁺. The optimization process comprised four distinct steps, which involved genome mining for insertion site variation, the optimization of linker length, linker content screening, and random mutagenesis of both the binding domain and the entire gene. ΔF/F rank plot representing all variants assessed during the directed evolution. For each round, tested variants are ranked from lowest to highest ΔF/F value from left to right. The red highlight dots are the candidates for validation in cultured mammalian cells. The underlying numerical data for this figure can be found in S1 Data.

53.37% (Fig 2B), which was 13-fold lower than that in the solution. We attribute this discrepancy to three primary factors: first, the intracellular environment may alter protein conformation, affecting sensor responsiveness to K⁺. Second, despite the addition of K⁺-specific ionophores like valinomycin and CCCP, ion permeation remains a challenge, resulting in lower effective ion concentrations interacting with the sensor in cells compared to direct in vitro assays. Third, in solution

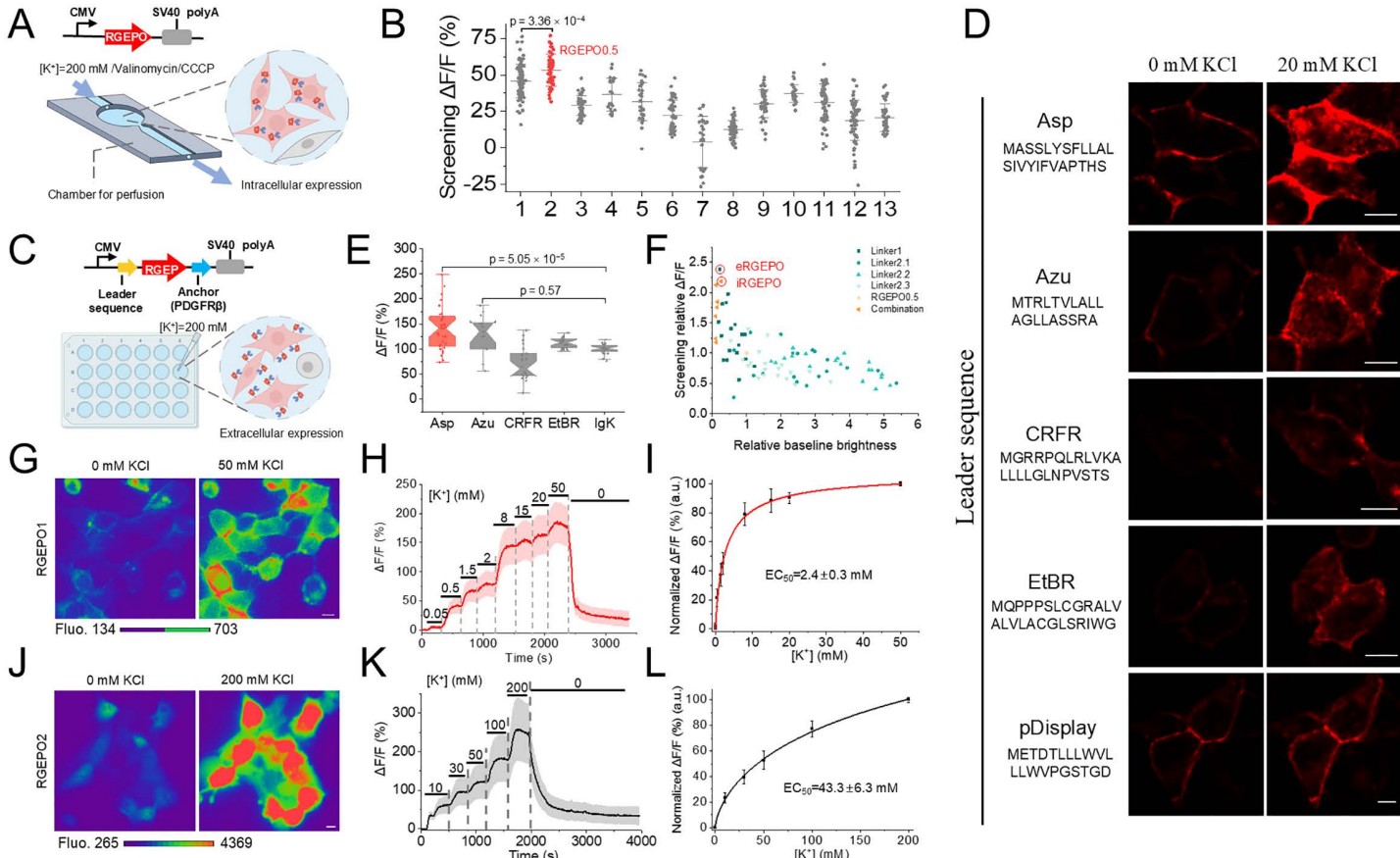

**Fig 2. Optimization in HEK293FT cells. (A)** The linear diagram of the expression cassette for RGEPO expression in the mammalian cells (top) and schematic representation of the automated perfusion system employed in buffer exchange experiments (created in BioRender.com). **(B)** Validation of candidates selected in *Escherichia coli* using HEK293FT cells via perfusing K$^+$ free and 200 mM K$^+$ containing buffers($n$ = 80, 50, 46, 22, 29, 50, 29, 54, 39, 26, 65, 66, 43 cells from 1 culture for 1st to 13th mutant). One-way ANOVA with Benferroni's test was performed. $p$ = 3.35775 × 10$^{-4}$ between mutant1 and mutant2. This screening led to RGEPO0.5. **(C)** A linear map of the expression cassette for RGEPO expression at the extracellular surface of the plasma membrane (top) and a schematic depiction of the detection system employed to assess the response of RGEPO (created in BioRender.com). **(D)** Representative single-plane confocal fluorescence images of leader-sequence variants ($n$ = 3 cells from 2 FOVs over 1 culture for each). Scale bars, 10 μm. **(E)** ΔF/F of a range of leader sequence variants in response to 200 mM K$^+$ ($n$ = 31, 13, 22, 22, and 24 cells from 1 culture for Asp, Azu, CRFR, EtbR, and Igκ, respectively). Box plots with notches are used: narrow part of notch, median; top and bottom of the notch, 95% confidence interval for the median; top and bottom horizontal lines, 25% and 75% percentiles for the data; whiskers extend 1.5 × the interquartile range from the 25th and 75th percentiles; horizontal line, mean; outliers, dots. One-way ANOVA with Benferroni's test was performed. $P$ = 5.0538 × 10$^{-5}$ between Asp and Igκ. $P$ = 0.05682 between Azu and Igκ. **(F)** Saturation mutagenesis screening and recombination of linker content on the extracellular surface of cultured mammalian cells with the Asp leader sequence, continuing from the template in **(B)**. This screening led to RGEPO1 and RGEPO2 indicated with a red circle. **(G)** Representative images of expression and fluorescence change of RGEPO1 in response to 20 mM K$^+$. Scale bar, 10 μm. **(H)** Fluorescence intensity change (ΔF/F) time course of RGEPO1 with stimulation by a series of K$^+$ buffers on HEK293FT cells ($n$ = 27 cells from 1 independent culture, see the other 2 independent culture data in the S9A and S9B Fig), data are expressed as mean ± SD. **(I)** Plot of normalized ΔF/F against different K$^+$ concentrations fitted using nonlinear fitting (Hill) for the data shown in panel H and S9A and S9B Fig ($n$ = 85 cells from 3 independent cultures). **(J)** Representative images of expression and fluorescence change of RGEPO2 in response to 200 mM K$^+$. Scale bar, 10 μm. **(K)** Fluorescence intensity change (ΔF/F) time course of RGEPO2 with stimulation by a series of K$^+$ buffer on HEK293FTZZ cells using 10 μg/mL gramicidin ($n$ = 29 cells from 1 independent culture, see the other 2 independent culture data in the S9C and S9D Fig), data are expressed as mean ± SD. **(L)** Plot of normalized ΔF/F against different K$^+$ concentrations fitted using nonlinear fitting (Hill) for the data shown in panel K and S9CC and S9D Fig ($n$ = 97 cells from 3 independent cultures). The underlying numerical data for this figure can be found in S1 Data.

titrations start from 0 mM, whereas intracellular conditions inherently contain potassium ions, affecting baseline fluorescence responses. These challenges necessitate further optimization to enhance $\Delta F/F$ in mammalian cells.

To enhance the screening efficiency in HEK293FT cells, we decided to express the sensor on the extracellular surface of the plasma membrane, which facilitates rapid buffer exchange to modulate sensor fluorescence and thus increase the throughput of the screening. Using RGEPO0.5 as a template for preliminary extracellular probe construction, we targeted it to the outer surface of HEK293FT cells by fusing it to five N-terminal leader sequences from naturally occurring proteins (see Methods for details) and the platelet-derived growth factor receptor (PDGFRβ) transmembrane domain, respectively (Fig 2C). While all generated constructs exhibited membrane localization and functional responses to K$^+$, the Asp leader peptide variant demonstrated a superior response, with a 1.4-fold higher $\Delta F/F$ compared to the widely used leader sequence of immunoglobulin k-chain (Igk) from pDisplay system [26] (Figs 2D, 2E, and S4). Additionally, compared to pDisplay-EGFP as a cell surface marker, the combination of Asp leader and PDGFRβ anchor also effectively targeted RGEPO0.5 to the cell surface (S5 Fig). Utilizing the Asp-PDGFRβ cell surface expression system, we performed saturation mutagenesis on the linker content. Stimulated with 200 mM K$^+$, the P299C mutant exhibited the highest response among the single and recombined linker mutations and was therefore designated as RGEPO1 (Fig 2F). Subsequently, we validated the top three mutants with the highest extracellular response by expressing them intracellularly and measuring fluorescence response through intracellular perfusion with 200 mM K$^+$ in the presence of 15 µM valinomycin and 5 µM CCCP. Among them, P152L/A298N showed the highest response to K$^+$ intracellularly and was thus designated as RGEPO2 (Figs 2F and S6). As a result, RGEPO1 and RGEPO2 had G264V mutations in the binding domain but differ only by linker content between each other (see S7 Fig for amino acid alignment of the prototype, RGEPO0.5, and RGEPO1,2).

To characterize the EC$_{50}$ and the dynamic range of RGEPOs and confirm the reversibility of the fluorescence response, we performed serial titrations with K$^+$ in HEK293FT cells. The Asp-RGEPO1 sensor exhibited clear localization to the cell membrane. The addition of K$^+$ triggered a robust fluorescence increase of cell surface-localized RGEPO1 (Fig 2G), which returned to baseline after washing out K$^+$ using a perfusion system (S8A Fig). The fluorescence increase of RGEPO1 was dependent on K$^+$ concentration via the perfusion system, achieving a $\Delta F/F$ of 180% at 50 mM (Figs 2G, 2H, S9A, and S9B; for visualization of fluorescence changes see S1 Movie). The EC$_{50}$ at the cell surface was determined to be 2.4 ± 0.3 mM enabling detection of [K$^+$] in the range of 0.5–20 mM, with a Hill coefficient of 0.9 ± 0.1 (Fig 2I). In parallel, RGEPO2 was observed to localize evenly within the cytoplasm and nucleus without any visible aggregates (S8B Fig). To better assess its responsiveness under intracellular conditions, we performed final titrations using gramicidin [27], which more effectively depletes intracellular K$^+$ and provides a clearer baseline compared to valinomycin used during earlier screening steps. This switch allowed for more accurate characterization of the sensor's dynamic properties in cells. Under gramicidin treatment, RGEPO2 exhibited a robust fluorescence response of 334% at 200 mM K$^+$, with an EC$_{50}$ of 43.3 mM and a Hill coefficient of 0.7 ± 0.1 (Figs 2J, 2K, 2L, S9C, and S9D; for visualization of fluorescence changes see S2 Movie). For comparison, valinomycin-based titrations, which were used during the optimization stage, yielded an EC$_{50}$ of 16.8 mM (S10 Fig), reflecting a higher intracellular K$^+$ baseline. Fluorescence returned to baseline after washout, confirming full reversibility of RGEPO2 in HEK293FT cells. As a result, we developed a pair of red GEPOs for detecting intracellular and extracellular K$^+$ dynamics in the wide range of potassium concentrations from 0.5 to 200 mM, with properties suitable for cellular imaging applications.

Given the potential application of RGEPOs in dual-color imaging experiments involving blue light, we also evaluated their susceptibility to photoactivation. Since many mApple-derived sensors are prone to photoactivation in the red channel under blue light [28–30], we assessed whether RGEPOs exhibited similar behavior. Using a protocol adapted from Hod Dana and colleagues [31]. for red calcium sensor characterization, we tested RGEPOs in HEK293FT cells by continuous imaging under green-light excitation (555/20 nm), combined with five additional blue-light stimulations (470/28 nm, 2.1 mW/mm$^2$, 5 s per pulse). As a result, we observed that RGEPOs exhibit a detectable degree of blue-light-induced photoactivation (S11 Fig), similar to that reported for jRGECO1a.

Building on these functional validations, we established a hybrid workflow for sensor development, combining high-throughput library screening in *E. coli* with subsequent optimization in HEK293FT cells. This pipeline enabled the efficient engineering of RGEPO variants based on the Hv-Kbp domain, resulting in two sensors, RGEPO1 and RGEPO2, with high dynamic range and full reversibility across physiologically relevant potassium concentrations.

## Characterization of RGEPOs in solution

To evaluate the applicability of the sensor in complex mammalian systems, it is essential to assess its fundamental biophysical properties in highly controlled environments, such as solutions. To this end, we performed a comprehensive characterization of the spectral and biochemical properties of RGEPO1 (without Asp leader or PDGFRβ anchor) and RGEPO2 in the physiological range of [$K^+$] in mammalian cells, i.e., from 0 to 150 mM [$K^+$] at pH 7.4 [25]. In a buffer without potassium, RGEPO1 and RGEPO2 exhibited two absorbance peaks at 448/446 and 576/578 nm, indicative of the protonated and the deprotonated chromophore forms, respectively (Fig 3A). Upon adding 150 mM $K^+$ to the purified proteins, we observed an increase in the absorption at 576/578 nm and a simultaneous absorption decrease at 448/446 nm (Fig 3A), suggesting a $K^+$ dependent chromophore deprotonation process. Correspondingly, excitation peaks were observed at 575 nm for RGEPO1 and 574 nm for RGEPO2, leading to emission maxima at 591 nm and 593 nm, respectively (Figs 3B and S12). In the $K^+$ binding state, the excitation maximum shifted slightly blue to 563 nm (S12 Fig), most likely due to the influence of the potassium ion on the chromophore environment. Upon the addition of 150 mM $K^+$, RGEPO1 emission showed a 4.7-fold intensiometric increase at its peak of 591 nm, and RGEPO2 exhibited a 3.1-fold at its peak of 593 nm (Fig 3B). The enhanced fluorescence intensity was attributed to increases in both effective extinction coefficients (EC) and

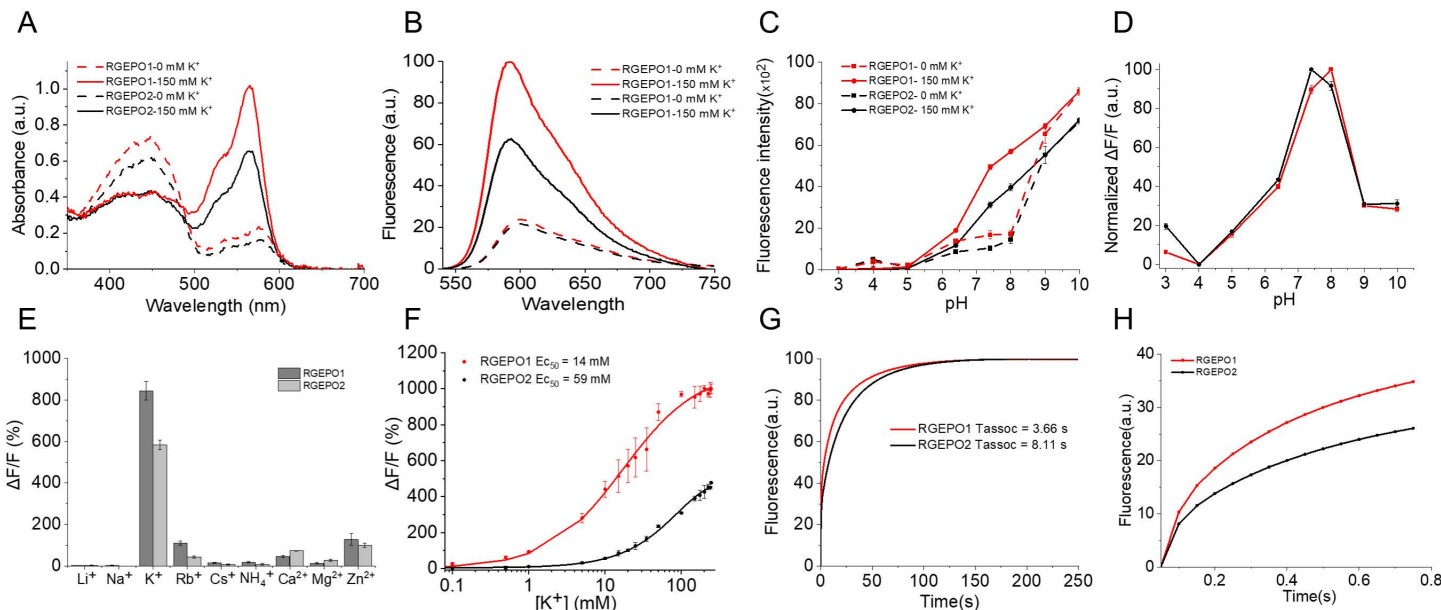

**Fig 3. In vitro characterization of RGEPOs. (A)** Absorbance spectra of RGEPOs at 0 and 150 mM potassium at pH = 7.4. **(B)** Steady-state fluorescence spectra of RGEPOs at 0 and 150 mM potassium at pH = 7.4. **(C)** Fluorescence intensity of RGEPOs at 0 and 150 mM potassium as a function of pH ($n = 3$ technical replicates; mean ± SD). **(D)** Normalized fluorescence changes upon the addition of 150 mM potassium as a function of pH. **(E)** Ion selectivity of RGEPOs ($n = 3$ technical replicates; mean ± SD). The concentrations of cations used were above their physiological concentrations. **(F)** $K^+$ titration of RGEPOs at isotonic conditions ($n = 3$ technical replicates for RGEPO1 and RGEPO2, respectively). **(G)** Association kinetics of RGEPOs at 150 mM $K^+$ measured by stopped-flow fluorimetry ($n = 3$ independent technical replicates each). **(H)** The same association kinetics curve as (G) shown in the range of the 0−0.75 s time frame. The underlying numerical data for this figure can be found in S1 Data.

quantum yields (Table 1). For RGEPO1, the effective extinction coefficient increased from $10{,}700 \pm 900$ to $41{,}600 \pm 500$ $M^{-1} \cdot cm^{-1}$, and for RGEPO2, it increased from $9{,}700 \pm 1{,}200$ to $26{,}850 \pm 2{,}960$ $M^{-1} \cdot cm^{-1}$. Concurrently, the quantum yields (QYs) under these conditions increased by ~1.7-fold for RGEPO1 and ~1.4-fold for RGEPO2 (Table 1). Since we intended to image RGEPOs in vivo using two-photon microscopy, we performed 2P fluorescence measurements of both RGEPO1 and RGEPO2 under $K^+$-free and $K^+$-saturated conditions (S13 Fig, S2 and S3 Tables). Both sensors exhibited robust potassium-dependent fluorescence under 2P excitation, supporting their suitability for two-photon imaging with optimal excitation wavelengths at 1,040 and 1,120 nm. For comparison, corresponding 1P measurements were conducted using the same samples under identical buffer conditions (S2 and S3 Tables). Together, these findings highlight the significant potential of RGEPO1 and RGEPO2 as effective indicators for monitoring potassium ion concentrations, attributed to their substantial changes in spectral properties upon $K^+$ binding. We also characterized the pH sensitivity of RGEPOs, which observed significant fluorescence changes in response to 150 mM $K^+$ within a pH range of 6.5–9.0. RGEPO1 achieved a maximum dynamic range at pH 8.0 and RGEPO2 at pH 7.4, with respective $pK_a$ values of 8.55 and 8.55 in

**Table 1. Spectral and biochemical properties of RGEPOs in solution.**

| Property | RGEPO1 | | RGEPO2 | |
|---|---|---|---|---|
| | 0 mM $K^+$ | 150 mM $K^+$ | 0 mM $K^+$ | 150 mM $K^+$ |
| Absorbance (nm) | 448/576 | 449/565 | 446 /578 | 449 /565 |
| Excitation (nm) | 575 | 564 | 574 | 563 |
| Emission (nm) | 599 | 593 | 599 | 591 |
| Extinction Coefficient ($M^{-1} \cdot cm^{-1}$) | 10,700 $\pm$900[a] | 41,600 $\pm$500[a] | 9,700 $\pm$1,200[a] | 26,850 $\pm$2,960[a] |
| Quantum Yield (%) | 14.6 | 24.6 | 16.2 | 22.2 |
| Molecular Brightness | 1,567 | 10,229 | 1,563 | 5,947 |
| Lifetime (ns) | 1.77 | 1.71 | 1.75 | 1.72 |
| $pK_a$ | 8.55$\pm$0.05[a] | 7.36$\pm$0.01[a] | 8.55$\pm$0.07[a] | 7.82$\pm$0.04[a] |
| $\Delta F/F_{max}$ in solution (%) | 844[b] | | 544[b] | |
| $EC_{50}$ in solution (mM) | 14[c] | | 59[c] | |
| $\Delta F/F_{max}$ in HEK293FT (%) | 180[d] | | 334[e] | |
| $EC_{50}$ in HEK293FT (mM) | 2.4$\pm$0.3[f] | | 43.3$\pm$6.3[g] | |
| $\Delta F/F_{max}$ in astrocyte (%) | 206[h] | | 430[i]/328[j] | |
| $K_d/EC_{50}$ in astrocyte (mM) | 4.4$\pm$1.1[k] | | 24.2$\pm$4.6[l]/50.4$\pm$6.9[m] | |
| $\tau_{assoc}$ (s) | 3.66[a,n] | | 8.11[a,n] | |

[a]Three independent technical replicates from one protein purification.

[b]Measured upon addition of $[K^+]=150$ mM under nonisotonic conditions in solution.

[c]Measured at the potassium concentration ranging from 0 to 245mM under isotonic conditions.

[d]Measured using local 50 mM KCl application via the perfusion system on the extracellular surface of HEK293FT cells.

[e]Measured via the perfusion system with 200 mM KCl in the cytoplasm of HEK293FT cells in the presence of gramicidin.

[f]Measued at the potassium concentration ranging from 0 to 50 mM on the extracellular surface of the HEK293FT cells.

[g]Measued at the potassium concentration ranging from 0 to 200 mM in the cytoplasm of HEK293FT cells in the presence of gramicidin.

[h]Measured using local 50 mM KCl application via the perfusion system on the extracellular surface of astrocytes.

[i]Measured via the perfusion system with 200 mM KCl in the presence of valinomycin and CCCP in the cytoplasm of astrocytes.

[j]Measured via the perfusion system with 200 mM KCl without valinomycin or CCCP in the cytoplasm of astrocytes.

[k]Measued at the potassium concentration ranging from 0 to 20 mM on the extracellular surface of astrocytes.

[l]Measued at the potassium concentration ranging from 0 to 200 mM in the plasma of astrocytes in the presence of valinomycin and CCCP.

[m]Measued at the potassium concentration ranging from 0 to 200 mM in the plasma of astrocytes without valinomycin and CCCP.

[n]Measured at $[K^+] =150$ mM under isotonic conditions in solution.

the K⁺-free state and p$K_a$ values of 7.36 and 7.82 in the K⁺-bound state for RGEPO1 and RGEPO2, respectively (Fig 3C and 3D). Thus, the RGEPOs' fluorescence was sensitive to physiologically relevant pH change. To investigate the metal ion selectivity, we titrated RGEPOs with a variety of ions: Li⁺, Na⁺, Rb⁺, Cs⁺, NH₄⁺, Mg²⁺, Ca²⁺, and Zn²⁺, along with K⁺. Among all tested ions, only Rb⁺, Ca²⁺, and Zn²⁺ induced considerable fluorescence increases of RGEPO1/RGEPO2 by 109.7%/43.6%, 45.5%/72.7%, and 127.1%/100.0%, respectively (Fig 3E). At the highest concentrations of the tested ions typically found in mammalian cells [31–38], ΔF/F for K⁺ were the highest at 844% for RGEPO1 and 584% for RGEPO2 significantly surpassing responses to other ions. Although Ca²⁺ appeared to cause moderate fluorescence increases, we conducted a refined titration of both RGEPO1 and RGEPO2 with Ca²⁺ ranging from 0 to 1.07 mM to assess its potential physiological interference. The results demonstrated that neither sensor responds to Ca²⁺ under physiological conditions (S14 Fig), confirming their selectivity for K⁺. Median effective concentrations (EC₅₀) were determined as 14 mM for RGEPO1 and 59 mM for RGEPO2 under isotonic conditions with varying K⁺ concentrations up to 245 mM (Fig 3F). The corresponding Hill coefficients were 0.88 ± 0.08 for RGEPO1 and 1.1 ± 0.07 for RGEPO2. According to these results, the estimated dynamic range of K⁺ detection in solution was from ~0.1 to ~225 mM. Overall, these results demonstrate the strong potential of RGEPOs as selective and sensitive indicators for potassium ions in varying physiological environments.

Potassium binding kinetics were further explored at 150 mM K⁺ using stopped-flow fluorimetry, which revealed an exponential association curve for both sensors. The components of this curve led to a 50% fluorescence change within approximately 3.66 s for RGEPO1 and 8.11 seconds for RGEPO2 (Fig 3G and 3H). Additionally, we measured fluorescence lifetimes. The average fluorescence lifetimes for RGEPO1 in the K⁺-free and K⁺-bound states were 1.77 and 1.71 ns, respectively. For RGEPO2, the lifetimes were 1.75 and 1.72 ns in the K⁺-free and K⁺-bound states, respectively (Table 1). Altogether, we characterized the fluorescence spectra, brightness, affinity, pH sensitivity, metal ion specificity, and kinetics of RGEPOs, demonstrating that RGEPO1 and RGEPO2 can serve as sensitive and selective indicators for monitoring K⁺ dynamics across various physiological environments.

## Molecular mechanism of potassium sensitivity

To explore the molecular mechanism underlying potassium binding, we performed classical molecular dynamics (MD) simulations for both the RGEPO1 and RGEPO2 systems in solution at low (20 mM) and high (300 mM) potassium concentrations. Low concentration was selected to match the affinity of the sensors in the solution, and high concentration corresponds to the upper limit of the sensor dynamic range. The initial protein structures were generated using AlphaFold2, revealing two key structural differences in the linker regions of RGEPO1 and RGEPO2 that were maintained and stable in the MD (S15 Fig). Specifically, in RGEPO2, residues N298 and P299 exhibit a tendency towards an open conformation, which results in the loss of beta-sheet character below residue P299 (S16 Fig). This structural variation renders the phenolate end of the chromophore in RGEPO2 more accessible to the solvent as compared to RGEPO1. This could explain some of the differences in molecular brightness in K⁺-bound states between them, since the fluorescence mechanism likely requires a structural shift upon K⁺ binding to stabilize the phenolate and emissive form. RGEPO2, therefore, would require a larger conformational change as compared to RGEPO1, though they have similar recognition mechanisms, as discussed further below. The MD simulations of RGEPO1 revealed two distinct binding pockets for K⁺ ions (Fig 4A). The first, referred to as the main binding pocket, was observed in both 20 and 300 mM ionic concentrations and exhibited significant K⁺ binding duration (~140 ns; Fig 4A upper inset). The second pocket, termed the transient binding pocket, showed a shorter binding time (~100 ns) and was observed only at 20 mM (Fig 4A lower inset). The main binding pocket was formed by 13 amino acids, including K240/E241/P242/V243/D294/L295/S297/A298/C299/V300/S301/V349/D350/R374. Among them, D294 and D350 directly coordinate potassium ions via ionic interactions. R374 is highly mobile prior to K⁺ binding, and subsequently forms a key hydrogen bond with D294 that stabilizes K⁺ in the pocket for a significant amount of time. The transient pocket consisted of only 5 amino acids, namely D156/V158/K159/S228/K229. Among them, only V158 and K159 were shared with the binding pocket identified in the Ec-Kbp domain of the KRaION1 biosensor [14].

In contrast, RGEPO2's MD simulations identified only one binding pocket under both ionic conditions (20 and 300 mM), in a similar location to the main binding pocket for RGEPO1, but all different amino acids in the binding pocket, which included W148/E149/A150/S151/D350/E352/K317 (Fig 4B). The binding region is located closer to the FP (Fig 4B). The bound ion remained stable for ~50 ns in this binding pocket. This positioning also caused an opening between two beta sheets near the phenolate O⁻ end, destabilizing the FP's beta-barrel structure and exposing the inner residues more to the solvent as compared to RGEPO1. This shorter timescale is consistent with the slightly lowered affinity of RGEPO2 observed experimentally in mammalian cells. The main K⁺-binding pockets are located either on the interface between the binding domain and FP in the case of RGEPO1 or in the β-barrel of FP next to the phenolate group of the chromophore. These binding pockets were not previously reported for any other single-FP-based biosensors that utilize other ion-binding domains. However, it was possible to engineer GFP-like FPs to have $Zn^{2+}$ or $Cu^{2+}$ binding pockets inside of the β-barrel, directly interacting with the chromophore [39].

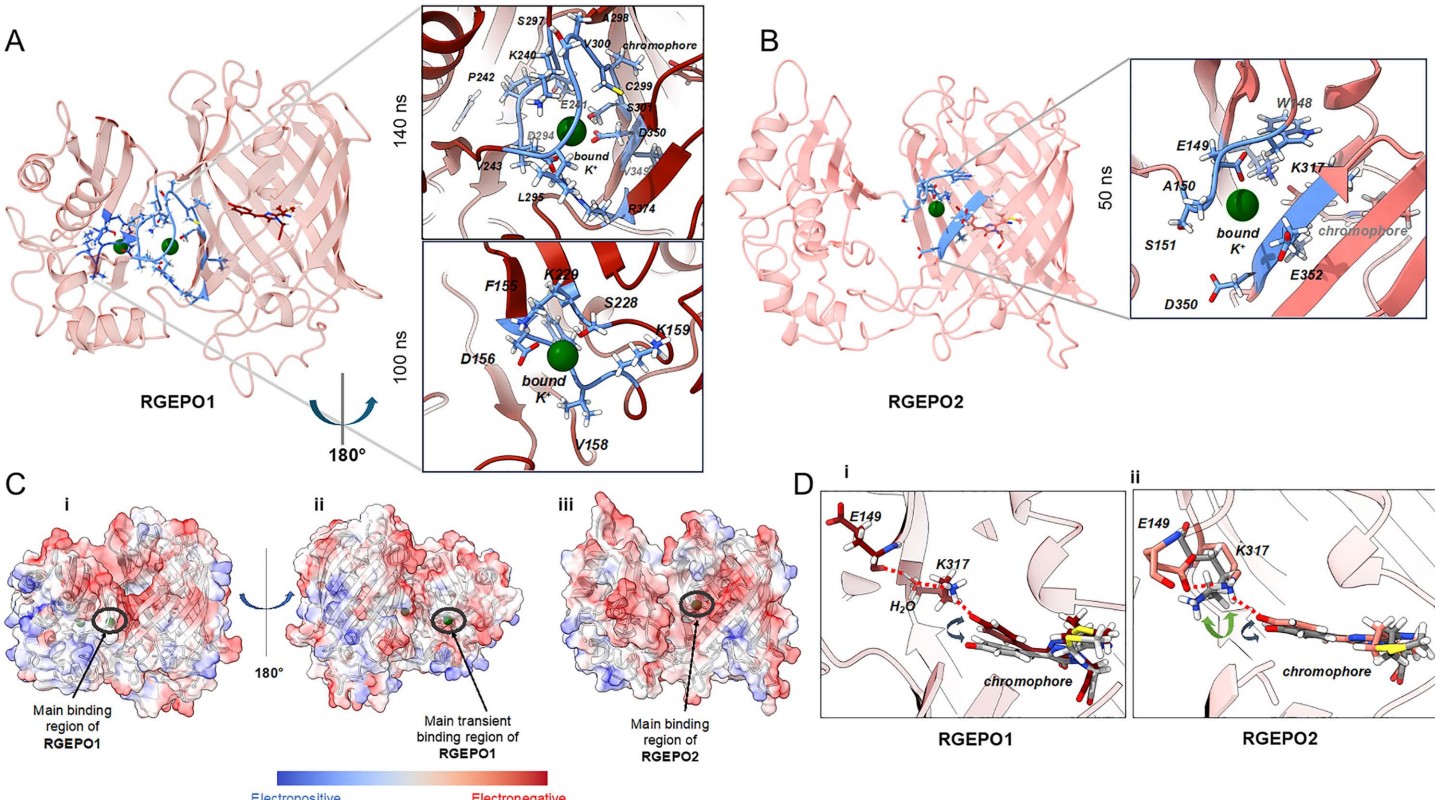

**Fig 4. Molecular mechanism of potassium sensitivity. (A)** The predicted 3D structure of RGEPO1 is shown in red, with K⁺ ions in green. The residues around the binding pockets are depicted as blue sticks. The binding regions are enlarged on the right with labels for the respective binding residues (upper main binding site and lower transient site). **(B)** The predicted 3D structure of RGEPO1 is shown in red, with K⁺ ions in green. The residues around the binding pockets are depicted as blue sticks. The binding region is enlarged on the right, and labels are placed for the respective binding residues. **(C)** Charge distribution of the surface of RGEPO1 **(i and ii)**, RGEPO2 **(iii)**. Surface color is assigned such that red is more electronegative and bluer is more electropositive. The green spheres represent the K⁺ ions. **(D)** Sensitivity mechanism of **(i)** RGEPO1 and **(ii)** RGEPO2. Left, the prominently observed hydrogen bonding network with O⁻ of phenolate ring of RGEPO1 upon K⁺ binding. Right, the prominently observed hydrogen bonding interactions with O⁻ of phenolate ring of RGEPO2 upon K⁺ binding. The hydrogen bonds are represented by thin red dashed lines, indicating the interaction points between the phenolate ring and surrounding residues/water (in sticks), and are labeled with their respective names. The chromophore's conformational space in both (i) and (ii) is depicted by gray sticks and double-headed arrows, indicating the range of movement in the absence of bound K⁺. In the right panel **(ii)**, the conformation of K317 in the absence of K⁺ is highlighted in gray sticks.

To further evaluate the ion selectivity of RGEPOs, we conducted MD simulations under high calcium conditions (300 mM $Ca^{2+}$) for both RGEPO1 and RGEPO2. Although $Ca^{2+}$ was occasionally observed near the chromophore region in both sensors, its binding was transient and substantially less stable than $K^+$. In RGEPO1, $Ca^{2+}$ localized to a solvent-exposed region adjacent to the transient $K^+$ binding pocket, but did not enter the main $K^+$ binding site (S17A–S17C Fig). The phenolate–K317 distance remained only briefly stabilized (~30–40 ns), and the surrounding water shell was not consistently displaced, indicating weak coordination (S17D–S17F Fig). In contrast, RGEPO2 allowed slightly deeper $Ca^{2+}$ entry into the chromophore-adjacent cavity (S17G–S17I Fig), but the binding remained unstable and short-lived. Notably, the H-bond network between $Ca^{2+}$, the chromophore, and K317 was far less persistent than that observed with $K^+$ (S17J–S17L Fig). Together, these results demonstrate that although $Ca^{2+}$ can transiently engage with surface-exposed acidic residues near the binding site, its interaction lacks the stability, depth, and configurational consequences required to activate the fluorescence response. This further confirms the selectivity of RGEPOs for $K^+$ over biologically relevant divalent cations such as $Ca^{2+}$.

To better understand the structural basis underlying this ion selectivity, we next examined the surface charge landscapes of RGEPO1 and RGEPO2. Further analysis of RGEPO1's surface charge distribution revealed an electronegative region extending from the main binding site to the transient site, indicating a higher likelihood of retaining positively charged ions. The MD shows $K^+$ ions transiently bound to this red region for a brief time (Figs 4C and S18). The more grooved areas, such as the main binding pocket and the transient binding regions, provide increased ion stability due to reduced solvent exposure, resulting in longer ion retention. Comparatively, RGEPO2's surface lacked a continuous negatively charged region. Instead, it had more negatively charged residues clustered around the main binding pocket, forming a groove that stabilized the cations in that specific area (Fig 4C). Collectively, these results suggest that RGEPO1 has a better performance as a fluorescent sensor than RGEPO2, which is consistent with our experimental results.

Even though RGEPO1 and RGEPO2 have completely different main binding pockets, further analysis of the MD results confirmed that both show the same mechanism related to fluorescence sensing. Upon binding of $K^+$ to the main binding pocket of both RGEPO1 and RGEPO2, we observed a chain of hydrogen bonding interactions occurring with the $O^-$ position of the phenolate ring of the chromophore. In RGEPO1, these interactions extend from the phenolate ring to K317 and through a water molecule to E149 (Figs 4D and S19). In RGEPO2, the same interactions occur without the solvent mediating the chain (Fig 4D). The H-bonding residence time between phenolate $O^-$ and K317 is approximately 80% and 40% of the trajectory for RGEPO1 and RGEPO2, respectively. The analysis of variation of distance between K317 at the sidechain N atom and chromophore phenolate $O^-$ atoms upon binding of $K^+$ to the proposed main and transient sites that the distance is lowered (<3 Å) and hydrogen bonding between K317 and $O^-$ of chromophore is more stabilized compared to unbound situations (S20 Fig). Additionally, analysis of the water shell around the phenolate $O^-$ end of all the systems shows a drop in the number of water molecules upon $K^+$ binding to the binding domain (S21 Fig).

To further investigate the binding regions, we examined the structure of the previously reported NMR-resolved $K^+$-binding site in Ec-Kbp of the KRaION1 biosensor (PDB ID: 7PVC). The electrostatic distribution of the binding site clearly differed from that of RGEPO1 and RGEPO2 (S22 Fig), suggesting that the binding regions were altered by the introduced mutations. Furthermore, the electronegative charge distribution introduced by mApple at the interface of the potassium binding domain and the FP compared to that of green FP (PDB ID: 7VCM) made the proposed binding sites more accessible and stable in RGEPO1 and RGEPO2. To determine if the previously reported Kbp binding pocket was still a possibility, we carried out MD simulations by placing the $K^+$ ion at this previously reported Kbp binding pocket. In RGEPO1, the bound ion remained stable in this region throughout the simulation, although another $K^+$ ion diffused freely to the proposed main binding pocket. The proposed transient pocket of RGEPO1 is located close to the previously reported site (~7 Å) (S22D Fig), but importantly neither induces the necessary conformational change in the chromophore and K317 (Fig 4D). Additionally, in RGEPO2, the $K^+$ binding at the previously reported site was unstable, lasting less than 20 ns. Analysis of the distance between the K317(NZ) atom and the $O^-$ atom of the chromophore shows no significant

difference between the bound and unbound states (S20F Fig). Taken together, this suggests that the observed main binding pockets in RGEPO1 and RGEPO2 are more likely to be involved in the fluorescence mechanism.

## Recording potassium transients in primary astrocyte and neuronal cultures

Potassium plays a crucial role in maintaining normal neuronal physiology [4,40,41]. Elevated extracellular concentrations of potassium can generate neuronal hyperexcitability [4]. For example, increased extracellular potassium chloride is extensively used to induce membrane depolarization in cultured neurons [42,43]. To investigate whether RGEPO1 could detect extracellular $K^+$ dynamics in neurons, in combination with GCaMP6f for intracellular calcium imaging, we expressed RGEPO1 on the extracellular side of the plasma membrane in primary hippocampal neurons using the Asp leading peptide, which displayed a clear membrane localization (Fig 5A; see S23 Fig for more representative images). Simultaneously, GCaMP6f was expressed in the cytoplasm (Fig 5A). With this acquisition setup, we next examined neuronal responses to extracellular potassium application. To avoid potential artifacts from blue light-induced photoactivation of RGEPOs, the red and green channels were acquired sequentially during dual-color imaging (see S11 Fig and Methods). Transient stimulation with 30 mM $K^+$ resulted in a robust increase of approximately 71% in RGEPO1 and 350% in GCaMP6f, with time to peak fluorescence of approximately 27 s and 3 s, respectively, both of which returned to baseline following $K^+$ washout (Fig 5B; see S24A and S24B Fig for two more representative traces). Previous studies have demonstrated that the excessive stimulation of glutamate receptors by glutamate causes a significant efflux of potassium ions from neurons [15,44,45]. To investigate whether these changes in intracellular $K^+$ could be optically detected using RGEPO2, we expressed RGEPO2 in primary mouse neurons, where it showed even distribution throughout soma and dendrites (Fig 5C; see S23 Fig for representative images). To further validate the potassium efflux in response to neuronal activation, we co-expressed RGEPO2 and GCaMP6f in the same hippocampal neurons (Fig 5C). Upon 500 μM glutamate stimulation, GCaMP6f showed a robust fluorescence increase of approximately 1,076%, with a time to peak of 9 s, while RGEPO2 exhibited a fluorescence decrease of around 48%, peaking at 72 s (Fig 5D; for visualization of fluorescence changes see S3 Movie). These results are consistent with previous findings using the FRET-based potassium sensor GEPII, which demonstrated a pronounced intracellular $K^+$ decrease upon NMDA receptor activation in primary cerebellar granule neurons [46]. Together, our findings highlight the potential of RGEPOs as promising tools for monitoring the transient dynamics of intracellular and extracellular potassium in cultured hippocampal neurons.

To explore the simultaneous imaging of $K^+$ and $Ca^{2+}$ dynamics in the cytoplasm, we co-expressed RGEPO2 and the green $Ca^{2+}$ indicator GCaMP6f [47] in cultured hippocampal neurons (Fig 5E). Brief exposure to 30 mM KCl resulted in a rapid increase in the green fluorescence of GCaMP6f by approximately 930% due to $Ca^{2+}$ influx induced by neuronal depolarization (Fig 5F green trace; see S24C and S24D for more representative traces and S4 Movie for visualization of fluorescence changes). Concurrently, the red fluorescence of RGEPO2 underwent a decrease of approximately 49% (Fig 5F gray trace; see S24C and S24D for more representative traces and S4 Movie for visualization of fluorescence changes), suggesting $K^+$ efflux.

To explore the potential of combining RGEPOs with a green fluorescent potassium sensor for simultaneous monitoring of extracellular and intracellular potassium dynamics, we co-expressed RGEPOs and GINKO2 in cultured hippocampal neurons (S25 Fig). Upon transient $K^+$ stimulation, RGEPO1 exhibited a robust fluorescence increase of approximately 99%, while GINKO2 showed a fluorescence decrease of about 43% (S25B Fig). In contrast, RGEPO2 displayed a fluorescence decrease of approximately 51%, which was accompanied by an about 50% reduction in GINKO2 fluorescence (S25D Fig). These results demonstrate that RGEPOs performance was comparable to the well-established green potassium sensor GINKO2 in primary neurons and they can be co-imaged in the same cells, enabling dual-color imaging and simultaneous monitoring of intracellular and extracellular potassium dynamics.

Compared to neurons, astrocytes are essential for regulating and maintaining stable extracellular potassium levels, ensuring the overall homeostasis of the central nervous system [5,48]. To evaluate the functionality of RGEPOs, we

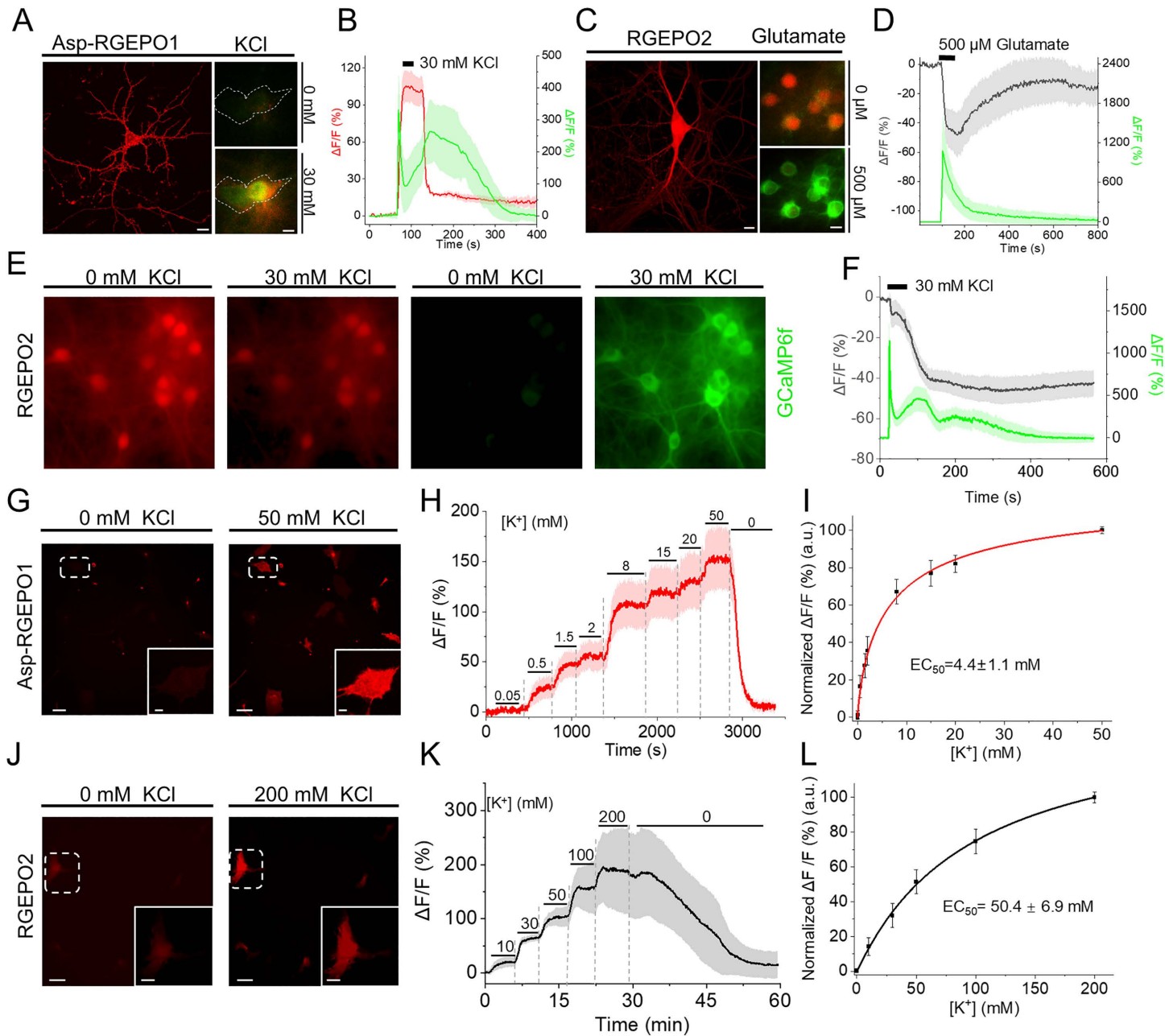

**Fig 5. Recording potassium transients in primary neuronal and astrocyte cultures. (A)** Representative image of hippocampal neurons co-expressing Asp-RGEPO1 with GCaMP6f, shown in ACSF buffer and in response to 30 mM KCl stimulation ($n=7$ neurons from 2 independent cultures; left, maximum intensity projection confocal image of Asp-RGEPO1, scale bar, 10 μm; right, wide-field images showing merged Asp-RGEPO1 and GCaMP6f fluorescence, scale bar, 10 μm). **(B)** Time course of fluorescence intensity changes of Asp-RGEPO1 and GCaMP6f with stimulation of 30 mM KCl on neurons ($n=5$ neurons from 1 independent culture, see the other 2 culture data in the S24A and S24B Fig), data are expressed as mean ± SD. **(C)** Representative image of a hippocampal neuron co-expressing RGEPO2 with GCaMP6f, shown in ACSF buffer and in response to 500 μM glutamate ($n=19$ neurons from 2 independent cultures; left, maximum intensity projection confocal image of RGEPO2, scale bar, 10 μm; right, wide-field images showing merged RGEPO2 and GCaMP6f fluorescence, scale bar, 10 μm). **(D)** Time course of fluorescence intensity changes of RGEPO2 and GCaMP6f with stimulation of 500 μM Glutamate ($n=98$ neurons from 3 independent cultures). **(E)** Representative dual-color wide-field imaging of K⁺ and Ca²⁺ dynamics in dissociated hippocampal neurons with response to 30 mM KCl. **(F)** Time course of fluorescence intensity changes of RGEPO2 and GCaMP6f with stimulation of 30 mM KCl in neurons ($n=31$ neurons from 1 independent culture, see the other 2 independent culture data in the S24C and S24D Fig), data are expressed as mean ± SD. **(G)** Representative wide-field images of primary astrocyte expressing RGEPO1 in response to 50 mM KCl ($n=8$ astrocytes from 1 independent culture). Scale bars, 100 μm, insert, 20 μm. **(H)** Time course of fluorescence intensity changes of RGEPO1

with stimulation by a series of KCl buffers on astrocytes ($n = 8$ astrocytes from 1 independent culture, see the other 2 independent culture data in the S27A–S27C Fig), data are expressed as mean ± SD. **(I)** Plot of normalized $\Delta F/F$ against different $K^+$ concentrations fitted using nonlinear fitting (Hill) for the data shown in panel H and S27A–S27C Fig ($n = 43$ astrocytes from 3 independent cultures). **(J)** Representative wide-field images of primary astrocyte expressing RGEPO2 in response to 200 mM KCl ($n = 5$ astrocytes from 1 independent culture). Scale bars, 100 μm, insert, 10 μm. **(K)** Time course of fluorescence intensity changes of RGEPO2 with stimulation by a series of KCl buffer ($n = 5$ astrocytes from 1 independent culture, see the other 2 independent culture data in the S27D and S27E Fig), data are expressed as mean ± SD. **(L)** Plot of normalized $\Delta F/F$ against different $K^+$ concentrations fitted using nonlinear fitting (Hill) for the data shown in panel J and S27D and S27E Fig ($n = 27$ astrocytes from 3 independent cultures). The underlying numerical data for this figure can be found in S1 Data.

transiently expressed them in primary mouse astrocytes (Fig 5G, 5H, and 5L). RGEPOs displayed clear localization on the extracellular membrane and in the cytoplasm of the astrocytes (S23 and S26 Figs). The fluorescence increase of extracellularly expressed RGEPO1 was dependent on $K^+$ concentration, achieving a $\Delta F/F$ of 206% at 50 mM (Fig 5G and 5H; see S27A–S27C Fig for more representative traces; for visualization of fluorescence changes see S5 Movie). The $EC_{50}$ at the cell surface was determined to be $4.4 \pm 1.1$ mM via the perfusion system with a Hill coefficient of $0.8 \pm 0.1$ (Fig 5I). Similarly, the fluorescence intensity changes of intracellularly localized RGEPO2 were also dependent on $K^+$ concentrations, resulting in a $\Delta F/F$ of 328% at 200 mM with an $EC_{50}$ of approximately $50 \pm 6.9$ mM and a Hill coefficient of $1.1 \pm 0.1$. (Fig 5J–5L; see S27D and S27E Fig for more representative traces; for visualization of fluorescence changes, see S6 Movie). The observed changes in RGEPO2 were consistent with a previous study using $K^+$ fluorescent dye [49]. Additionally, we investigated whether ionophores could also enhance potassium uptake in astrocytes. In the presence of valinomycin and CCCP, the fluorescence changes of RGEPO2 increased to 430% with an $EC_{50}$ of approximately 24 mM and a Hill coefficient of $0.9 \pm 0.1$ (S28 Fig). The observed difference in affinities under these two conditions can be attributed to the role of valinomycin in facilitating $K^+$ influx. This explains why, at the same titration concentrations, the valinomycin-treated group exhibited a larger fluorescence increase, particularly at the initial titration stages, resulting in higher affinity. Moreover, compared to HEK293FT cells, the fluorescence change in astrocytes was 3-fold larger at 200 mM $K^+$ in the presence of valinomycin and CCCP. This further demonstrates the superior ability of astrocytes to regulate potassium ion concentrations. Collectively, these results suggest that elevated extracellular $K^+$ concentration triggers $K^+$ uptake by astrocytes, supporting the notion that astrocytes play an important role in involving potassium spatial buffering and thus exhibit differential responses to potassium relative to neurons under similar depolarized conditions.

## Recording potassium transients in neurons in acute brain slices

Encouraged by the high performance of RGEPOs in cultured neurons, we extended our investigation to intact brain tissue. To simultaneously monitor $K^+$ dynamics and neuronal activity, we co-expressed Asp-RGEPO1 or RGEPO2 with GCaMP6f in the cerebral cortex neurons of P0 mice, using recombinant adeno-associated virus (rAAV2/9) driven by the human synapsin 1 (hSyn) promoter (Fig 6A). After 3–4 weeks, we prepared acute brain slices and imaged the fluorescence responses to application of extracellular $K^+$ (Fig 6A). Bright fluorescence was detected in the cortex region of the brain slices, confirming the expression of RGEPO1 or RGEPO2 and GCaMP6f (Figs 6B, 6C, S29, and S30). Stimulation with 20/30 mM KCl elicited a robust and repeatable fluorescence increase of 6.5% for RGEPO1 and 104% for GCaMP6f, both of which returned to baseline after KCl washout (Fig 6D–6F). Conversely, the application of 20/30 mM KCl led to a considerable decrease in RGEPO2 fluorescence by 7.0%, accompanied by a corresponding increase in the GCaMP6f signal of 128% (Fig 6G–6I). These trends were consistent with the results obtained from cultured neurons, although the $\Delta F/F$ values observed in acute brain slices were smaller. This discrepancy is likely due to differences in protein expression levels and the more complex regulatory networks present in intact tissue. Overall, these findings demonstrate that RGEPOs are capable of effectively detecting $K^+$ dynamics in both extracellular and intracellular environments in intact brain tissue.

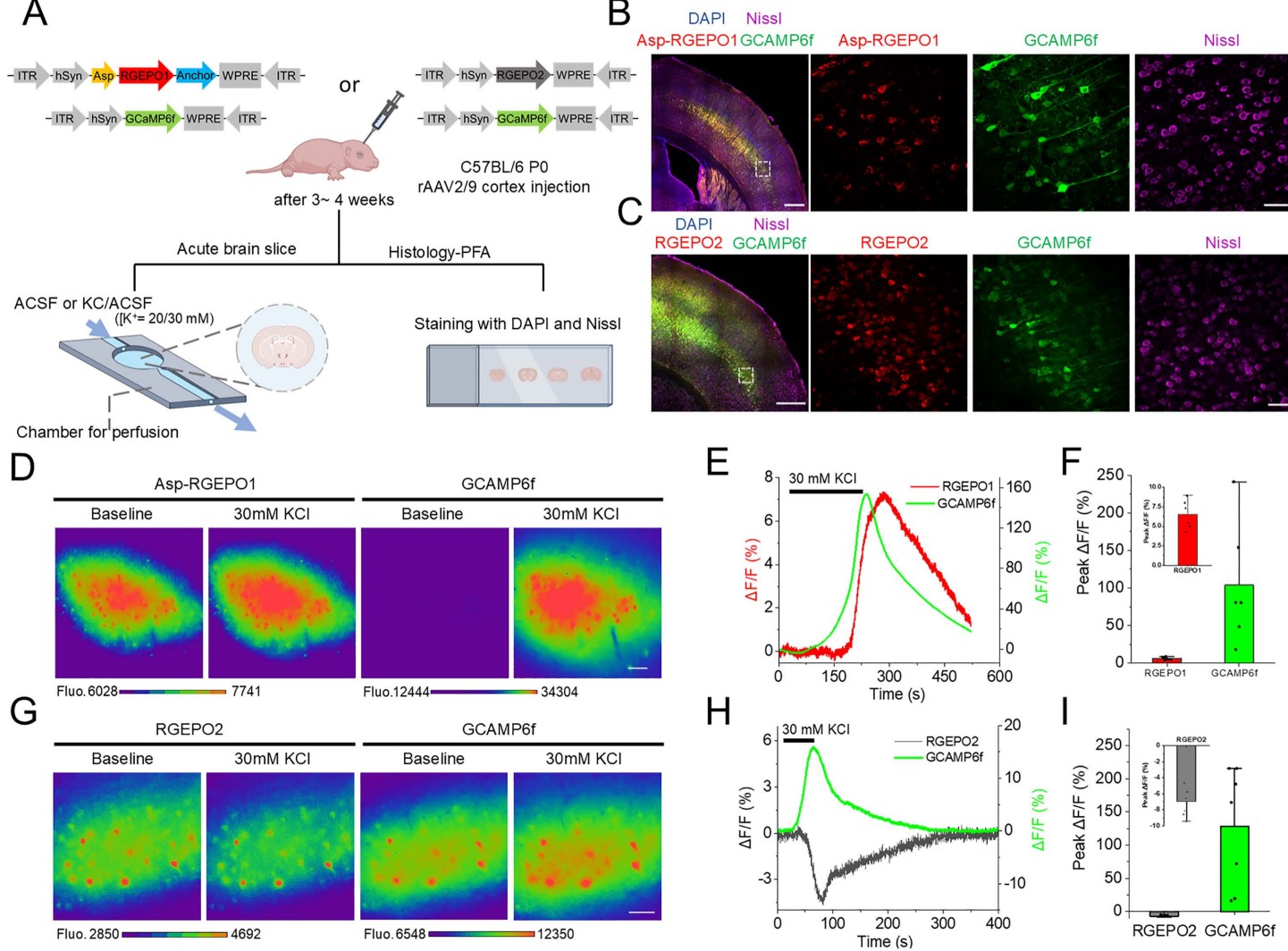

**Fig 6. Imaging of RGEPOs in neurons in acute brain slices. (A)** Schematic illustration depicting the AAV injection of Asp-RGEPO1 or RGEPO2 with GCAMP6f into the cortex for the brain slice experiments (created in BioRender.com). ITR inverted terminal repeat, hSyn human Synapsin I promoter, WPRE woodchuck hepatitis virus posttranslational regulatory element. **(B)** Left, fluorescence image of the cerebral cortex showing the expression of Asp-RGEPO1 (red) and GCAMP6f (green). Scale bar, 500 μm. Right, large magnification of fluorescence images showing relative expression regions. Scale bar, 50 μm. **(C)** Left, fluorescence image of the cerebral cortex showing the expression of RGEPO2 (red) and GCAMP6f (green). Scale bar, 200 μm. Right, large magnification of fluorescence images showing relative expression regions. Scale bar, 50 μm. **(D)** Representative pseudocolor images of RGEPO1 and GCAMP6f expressed in the brain slice in response to 30 mM KCl ($n = 4$ slices from 2 mice). Color bars, fluorescence intensity. Scale bar, 50 μm. **(E)** Single-trial optical traces of Asp-RGEPO1 and GCAMP6f fluorescence upon stimulation of 30 mM KCl of acute brain slice. **(F)** Quantification maximal ΔF/F of Asp-RGEPO1 and GCAMP6f in response to 20/30 mM KCl ($n = 6$ slices from 4 mice). Dot, individual data point for each acute brain slice; bar, mean; error bar, SD. **(G)** Representative pseudocolor images of RGEPO2 and GCAMP6f expressed in the brain slice in response to 30 mM KCl ($n = 2$ slices from 2 mice). Color bars, fluorescence intensity. Scale bar, 50 μm. **(H)** Single-trial optical traces of RGEPO2 and GCAMP6f in response to 20/30 mM KCl. **(I)** Quantification maximal ΔF/F of RGEPO2 and GCAMP6f in response to 30 mM KCl ($n = 7$ slices from 5 mice). Dot, individual data point for each acute brain slice; bar, mean; error bar, SD. The underlying numerical data for this figure can be found in S1 Data.

## Dual color neocortical imaging of Ca²⁺ and K⁺ dynamics during seizure

Epilepsy is a severe neurological disorder marked by defects in potassium channels and disrupted potassium homeostasis, leading to brain hyperexcitability and seizures [50,51]. Previous research using ion-selective probes has documented changes in potassium levels during seizures, but these methods are not scalable to monitor neuronal populations [52–55]. Building on the reliable detection of neuronal activity indicated by the potassium sensors in acute brain slices, we next explored the potential of monitoring potassium dynamics in neuronal populations using Asp-RGEPO1 and RGEPO2 in vivo in mice. In this experiment, we co-expressed GCaMP6f with RGEPO variants in neurons of the primary somatosensory cortex and employed two-photon microscopy to capture seizure activities in L2/3 neurons elicited by kainic acid (Fig 7A and 7B). Similar to previous findings [56–58], we observed a synchronous fluorescence wave followed by a propagating wave across the board in the case of calcium (Fig 7C–7E; see S31 Fig for additional traces of spreading waves recorded in other mice; S7 and S8 movies), which should indicate the stage of seizure activity and spreading wave of depolarization, respectively. In mice expressing Asp-RGEPO1, we detected a spreading wave of fluorescence increase (median peak $\Delta F/F = 89\%$) following seizure termination (Fig 7C and S7 Movie). The wave pattern was spatially consistent with the depolarization-driven calcium wave recorded by GCaMP6f, but exhibited a broader wavefront across neurons in the cortex (Fig 7D and 7G). The broader wavefront observed for RGEPO1 likely reflects the slower kinetics of the sensor relative to GCaMP6f, resulting in a more gradual fluorescence response pattern across neurons. Together, these results demonstrate that RGEPO1 can reliably report seizure-associated extracellular potassium dynamics in vivo and support its compatibility for dual-color imaging alongside the rapid response of calcium indicators. For RGEPO2, we observed a rapid fluorescence decay (median peak $\Delta F/F = -52\%$) coinciding with the seizure onset wave seen in GCaMP6f (median peak $\Delta F/F$ 136%; Fig 7E). RGEPO2 also exhibited a subsequent spreading wave (Fig 7E, 7F, and 7H). The RGEPO2 wave propagates within a similar time frame to GCaMP6f, as evidenced by approximate onset of the signal (Fig 7E and 7H and S8 Movie). These findings indicate a decrease in intracellular potassium levels reported by RGEPO2 during seizures, which is consistent with the study using the GEPII sensor to monitor kainic acid-induced K⁺ dynamics in cultured hippocampal neurons [59]. Compared to the results obtained from the acute brain slice data, RGEPOs in the mouse seizure model exhibited similar trends but produced stronger responses. Overall, these results demonstrated the capability of RGEPO1 and RGEPO2 to report potassium activity during seizures in vivo in mice. Although they offer lower temporal resolution and signal amplitude compared to GCaMP6f, RGEPO1, and RGEPO2, they enable the observation of potassium dynamics with single-cell resolution at the population level in vivo in drug-induced seizures for the first time.

## Discussion

In this study, we engineered the first red genetically encoded fluorescent potassium indicators, RGEPO1 and RGEPO2, by incorporating the Hv-Kbp, a novel potassium-binding domain, into the mApple FP (S7 Fig). Compared to green fluorescent potassium sensors [13–15] (S1 Table), RGEPOs offer several advantages, including compatibility with multicolor imaging and low phototoxicity and the ability to be applied in intact tissue environments. Additionally, RGEPOs exhibited a sensitive response to the intracellular and extracellular K⁺ change in the culture neurons and astrocytes, acute brain slices, as well as in awake mice. Our results collectively suggest that RGEPOs are effective tools for detecting intracellular and extracellular potassium during the neuronal activity process.

To engineer RGEPOs, we inserted the potassium binding domain into the red FP mApple at position 151 on the 7th β-strand next to the chromophore (Fig 1C). Majority of the previously reported red genetically encoded fluorescent probes have been based on the fusion of circularly permuted fluorescent protein with binding domain [60,61]. A similar molecular design has been used to develop glutamate and calcium biosensors, which were also constructed by inserting corresponding binding domains into the 7th β-strand of mApple [20,62]. However, the applicability of these biosensors was not demonstrated beyond cultured mammalian cells. In contrast, RGEPOs displayed a strong response in the cultured neurons and

PLOS Biology

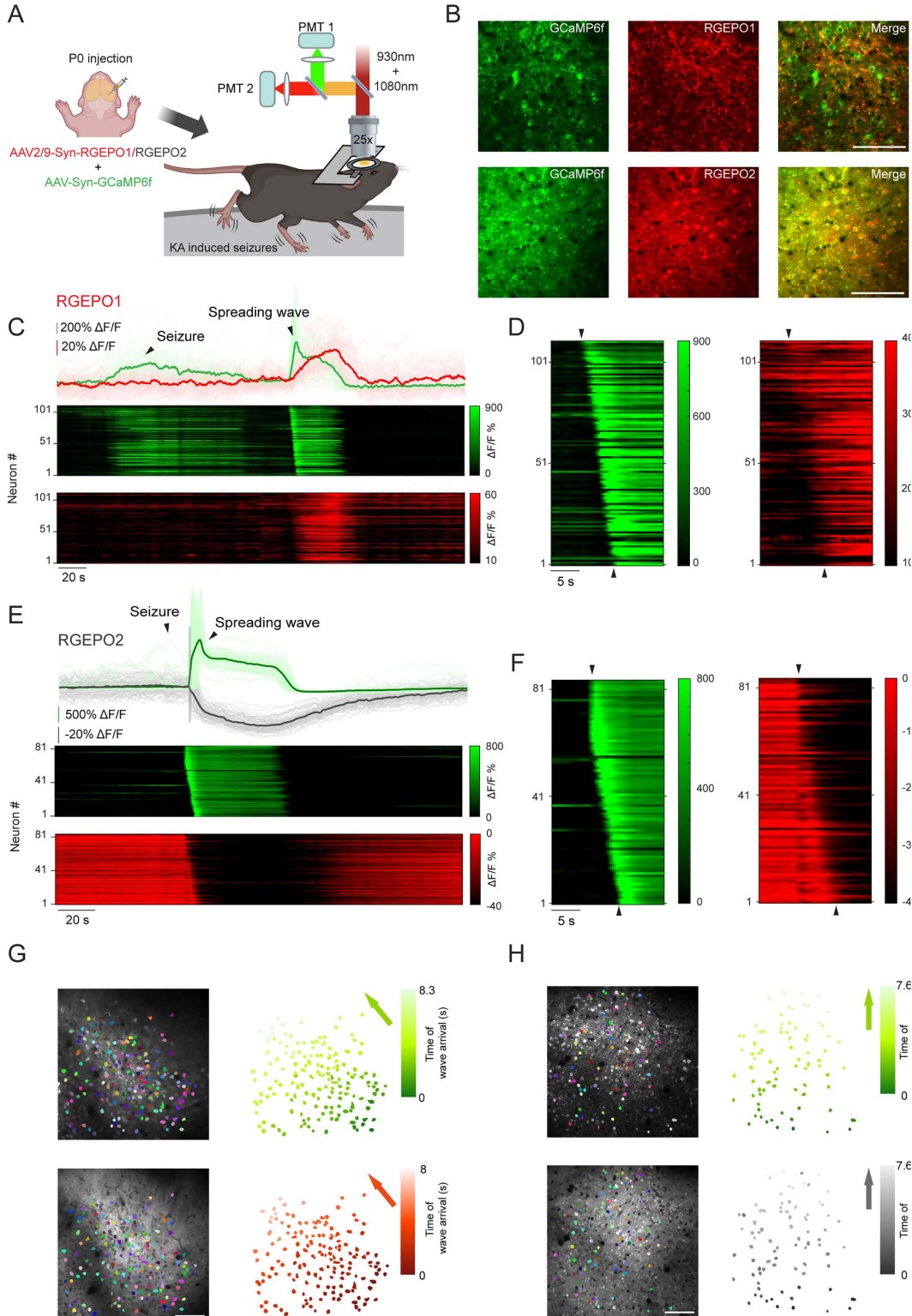

**Fig 7. Two-photon imaging of GCaMP6f and RGEPOs in mice with KA-induced seizures. (A)** Schematic representation of virus injection and imaging in a mouse undergoing KA-induced seizures using a two-photon microscopy (created in BioRender.com). **(B)** Representative single-plane images showing co-expression of RGEPO1/2 and GCaMP6f in layer 2/3 neurons of the primary somatosensory cortex. **(C)** Top panel: Representative individual

(light green and light red) and averaged (green and red) fluorescence responses of GCaMP6f and RGEPO1 during KA-induced seizures and subsequent spreading waves. Middle and bottom panels: Vertical projections of fluorescence profiles from neurons expressing GCaMP6f and RGEPO1 during KA-induced seizures and subsequent spreading waves. Data analysis from one mouse, see the other mouse data in the S31D Fig. Zoomed-in view of the vertical line profile of GCaMP6f and RGEPO1 from **(C)**, highlighting the onset of the spreading wave. $n = 111$ neurons from one mouse, replicated in 2 mice. **(E)** Top panel: Representative individual (light green and light red) and averaged (gray and dark gray) fluorescence responses of GCaMP6f and RGEPO2 during KA-induced seizures and subsequent spreading waves. Middle and bottom panels: Vertical projections of fluorescence profiles from neurons expressing GCaMP6f and RGEPO2 during KA-induced seizures and subsequent spreading waves. The gray shaded box indicates the period of acute motion observed along the z-axis. Data analysis from one mouse, see more mice data in the S31 Fig. **(F)** Similar to panel D, but for GCaMP6f and RGEPO2. $n = 83$ neurons from one mouse, replicated in 3 mice. **(G)** Temporal representation of the spreading wave of GCaMP6f and RGEPO1 following seizure activity. Left: pseudo-colored mask of individual neurons overlaid on the average projection image. Right: same mask as left, colored based on fluorescence peak time. $n = 160$ neurons from one mouse, replicated in 3 mice. **(H)** Similar to panel G, but for GCaMP6f and RGEPO2. $n = 83$ neurons from one mouse, replicated in 3 mice. The gray shaded box indicates the period of acute motion observed along the z-axis. Colored arrow indicates the estimated direction of wave propagation. Scale bar: 100 μm. The underlying numerical data for this figure can be found in S1 Data.

astrocytes (Fig 5) and were also capable of detecting K⁺ dynamics in both extracellular and intracellular environments of intact brain tissue (Fig 6). This opens new avenues for the development of red fluorescent probes, making RGEPOs a successful example of building red fluorescence protein-based indicators using the inserted-into-red FP topology.

RGEPOs were optimized through a combination of directed molecular evolution in *E. coli* and mammalian cells (Figs 1 and 2). This dual-system approach addresses the common challenge of transferring biosensor prototypes optimized in *E. coli* to mammalian cells [63], where they often perform suboptimally. By initially using the *E. coli* system combined with random mutant generation for high-throughput screening, we were able to identify promising prototypes, which were then validated in mammalian cells. Compared to optimization in a single system alone, this combined optimization strategy increases the throughput of sensor screening while overcoming the issue of prototype incompatibility between systems, thereby enhancing overall efficiency.

RGEPOs were successfully employed to detect intracellular and extracellular potassium dynamics in cultured neurons and astrocytes, as well as in acute brain slices. RGEPO1 exhibited sensitive responses to extracellular K⁺ across different models. RGEPO2 showed a negative response in primary neurons and a positive response in astrocytes, consistent with their physiological functions [4,5,48] (Fig 5). Furthermore, we observed that cultured neurons took longer to return to baseline after washing out K⁺ compared to astrocytes (Fig 5F and 5K), which support that astrocytes have a stronger capacity to regulate potassium ions [5,48]. This finding aligns with the understanding that while both neurons and astrocytes depend on K⁺ channels, the types and roles of these channels differ significantly [40,64]. In acute brain slices, we observed a similar response pattern to that seen in cultured neurons, though the response amplitudes were less pronounced, indicating the more complex K⁺ dynamics in the actual physiological environment.

To further extend the application of our potassium probes, we examined RGEPOs in vivo in mice using two-photon imaging. In mice with kainic acid (KA)-induced seizures, we observed a spreading wave in both RGEPO1 and RGEPO2, with time of wave arrival at cells observed across the board (Fig 7C–7H; S7 and S8 Movies). Specifically, Asp-RGEPO1 exhibited a propagating fluorescence increase that was spatially aligned with the calcium wave reported by GCaMP6f, consistent with cortical spreading depolarization. Compared to the sharp onset of GCaMP6f signals, the RGEPO1 wave exhibited a broader activation profile across neurons, likely reflecting the slower response kinetics of the potassium sensor. RGEPO2, by contrast, showed a rapid fluorescence decrease (median peak $\Delta F/F = -52\%$) that coincided with the seizure onset (Fig 7E), suggesting a drop in intracellular K⁺ levels. This was followed by a subsequent wave corresponding to the spreading depolarization. These dynamics are consistent with the known ionic shifts during seizure activity, where intracellular potassium efflux and calcium influx drive significant rises in extracellular K⁺ concentration and drops in intracellular K⁺ levels [65–67]. In summary, our results demonstrated the capability of RGEPO1 and RGEPO2 to capture key physiological changes in potassium levels, which, to our knowledge, is the first optical report of potassium transients at single-cell resolution during seizure-related activities in vivo in mice. Together, they represent a robust dual-color optical toolkit for studying K⁺ homeostasis in intact brain tissue.

Through MD simulations, we gained insights into the molecular mechanisms underlying potassium binding within RGEPO1 and RGEPO2. Our simulations revealed distinct potassium-binding pockets in the two indicators. RGEPO1 was found to have two potassium-binding pockets: a main binding pocket and a transient pocket similar to the potassium-binding site in Ec-Kbp. RGEPO2 has one potassium-binding pocket, which is close to the chromophore. These findings suggest that linker content changes can impact substrate binding, especially in proteins with simpler structures. Furthermore, the MD simulations also indicated that RGEPO1's surface is more electronegative, facilitating the retention of positively charged ions and supporting the notion that RGEPO1 performs better as a fluorescent sensor than RGEPO2, consistent with our solution-based results. Importantly, the simulations also confirmed the ion selectivity of RGEPOs, as $Ca^{2+}$ binding was found to be transient and insufficient to induce the conformational changes required for fluorescence activation. Together, these simulations provide mechanistic evidence supporting the potassium selectivity and sensing performance of RGEPOs.

The primary limitations of RGEPOs lie in their relatively low dynamic response range and limited fluorescence change ($\Delta F/F$) within intracellular environments, particularly in intact brain tissue. This issue was notably pronounced with RGEPO2, which exhibited lower $\Delta F/F$ in HEK293FT cells and neurons compared to green fluorescent potassium indicators. A key factor contributing to this reduced performance is likely the high basal potassium concentration in the cytoplasm, which attenuates sensor sensitivity. We also observed photoactivation with blue light, which is typical for mApple-derived red calcium biosensors [28,60], which would complicate combinations of RGEPOs with blue-light-driven optogenetic tools. To address these limitations, in our future studies, we plan to develop RGEPO variants with greater sensitivity, optimized binding affinity for improving the dynamics range, and minimal photoactivation with blue light. Moreover, we are working on engineering next-generation RGEPO variants with faster $K^+$ kinetics, enhancing their ability to detect $K^+$ pulses more effectively. Furthermore, we also observed puncta formation at high expression levels of RGEPOs, which we aim to mitigate by optimizing expression levels and binding domains. Finally, future efforts will focus on two-photon imaging experiments in awake mice to track cellular and subcellular $K^+$ dynamics under physiological and pathological conditions.

In summary, we have developed the first red genetically encoded indicators for imaging the intracellular and extracellular potassium dynamic in cultured mammalian cells, acute brain slices, and mice in vivo. RGEPOs represent significant advancements in the field of potassium imaging, offering new tools for the real-time visualization of $K^+$ dynamics in both extracellular and intracellular environments. Future work will focus on enhancing their sensitivity and expanding their applicability to other biological systems, ultimately contributing to a deeper understanding of potassium's role in cellular physiology.

## Methods

### Genome mining for the Ec-Kbp homologous domains

The BLASTp tool from the National Center for Biotechnology Information (NCBI) was utilized to identify alternative potassium-binding domains. We initiated our search with the amino acid sequence of Hv-Kbp (NCBI Sequence ID: VAV91021.1) as the query, targeting the env_nr metagenomic database. The search was executed with default parameters aside from the specified database. The acquired results were filtered based on the percentage similarity of amino acid sequences (above 40%), and manual selection was applied to refine the results. The following proteins were selected for further investigation: VAV (GenBank: VAV99455.1; hypothetical protein from hydrothermal vent metagenome); GAI (GenBank: GAI29416.1; unnamed protein product from marine sediment metagenome); MND (GenBank: MND59673.1; LysM domain/BON superfamily protein from compost metagenome); MPM (GenBank: MPM71561; putative protein YgaU from bioreactor metagenome); MNT (GenBank: MNT23082.1; LysM domain/BON superfamily protein from compost metagenome).

### Molecular cloning and mutagenesis

The DNA sequences encoding identified potassium binding domains, MNT (list all of them; sequence IDs above), and KRaION1 were synthesized by Tsingke Biotech Co. (China) and swapped with Electra1 gene in the pHybrid expression

vector [68] (pHybrid-Electra1, WeKwikGene plasmid #0000202) for screening in *E.coli* bacteria and cytoplasmic expression in HEK293FT cells. The secretion peptides used in Fig 2D were synthesized de novo with mammalian codon optimization based on the amino acid sequences of the N-termini of aspartic proteinase nepenthesin-1 (Asp, 1-24 aa from Sequence ID: BAF98915.1), azurocidin (Azu, 1-19 aa from Sequence ID: NP_001691.1), corticotropin-releasing factor receptor 1 (CRFR, 1-24 aa from Sequence ID: XP_040124593.1), endothelin receptor type B (EtBR, 1-26 aa from Sequence ID: CAD30645.1). For extracellular-cell-surface expression in HEK293FT cells, the corresponding genes were cloned and swapped with the Thyone gene in the pDisplay-Thyone plasmid (WeKwikGene plasmid #0000366). For generating prototypes of red potassium indicators, the genes of the selected potassium-binding domains were swapped with the gene of the CaM-M13 calcium-binding domain in the FRCaMPi gene, preserving the original linkers between FP and binding domains. For expression in primary mouse cell cultures, the RGEPOs genes were swapped with iRFP670-P2A-GFP in the pAAV-CAG-iRFP670-P2A-GFP plasmid (WeKwikGene plasmid# 0000008) and with GRAB_5-HT1.0 in the pAAV-hSyn-GRAB_5-HT1.0 plasmid (WeKwikGene plasmid# 0000123). To generate a membrane-targeting marker for astrocytes, the Igκ-EGFP-PDGFRβ sequence was used to replace the RGEPO1 gene in the pAAV-CAG-RGEPO1 plasmid (constructed in the previous step). All DNA fragments and plasmids were sequenced and verified by Healthy Creatures Biotec Co. (China).

The custom DNA oligonucleotides were synthesized by Tsingke Biotechnology Co., PrimeStar Max master mix (Takara, Japan) was used for routine polymerase chain reaction (PCR) amplification. The routine Gibson assembly was conducted using Hieff Clone Plus One-step cloning kit (YEASEN Biotech, China). Site-directed mutagenesis of RGEPOs was performed with overlap extension PCR using custom DNA oligonucleotides with degenerated NNS codons. Random mutagenesis was performed using error-prone PCR (GENSTAR, China) with the mutation rate 1–6 bp per kbp according to the manufacturer's instructions. Following purification on a 1% agarose gel, PCR products obtained using site-directed or random mutagenesis were enzymatically digested with KpnI and XbaI (New England BioLabs, USA), and ligated with a similarly digested pHybrid backbone vector via T4 DNA ligase (New England BioLabs, USA). Electroporation facilitated the introduction of ligation products into electrocompetent NEB 10-beta cells ($10^{10}$ cfu/µg; Biomed, China), which were then cultured in LB medium supplemented with 0.1 mg/ml ampicillin (Amp+) and 0.06% w/v L-Rhamnose (Rha+) (Sangon Biotech Co., China) for 24 h at 37 °C followed by culturing for 18 h at 18 °C. The BD FACS Melody Cell Sorter (BD Biosciences, USA) was utilized for bacterial library sorting, employing 561 nm excitation wavelength and 585/42 nm filter, from which approximately 30,000 cells with the highest fluorescence were sorted and cultured overnight on Amp+/Rha+ agar plates at 37 °C. Selected bright colonies were subsequently cultured in duplicates of 24 deep-well RB blocks (Invitrogen, USA), utilizing 4 mL of LB medium supplemented with ampicillin for plasmid extraction and another 4 mL of LB medium supplemented with ampicillin and rhamnose for protein expression. The cultures were initially incubated at 37 °C for 24 h, followed by a lower temperature incubation at 18 °C for 18 h. Post three freeze-thaw cycles at −80 °C to disrupt the cells, proteins were extracted using the Bacterial Protein Extraction Kit (Sangon Bitech, China). The lysates, titrated with KCl solution in the range of 0–200 mM, were assessed for fluorescence response in 96-well plates using the Varioskan LUX Plate reader (ThermoFisher Scientific, USA) with excitation at 560 nm and emission at 580–640 nm. Mutants, selected based on the highest fluorescence changes ($\Delta F/F$) and relative baseline brightness, were further sequenced and revalidated as described above.

## Protein purification and in vitro characterization

Protein expression and purification were carried out utilizing the rhamnose-inducible pHybrid system in TOP10 cells (Biomed, China) cultured in Amp+/Rha+ LB medium at 37 ℃ overnight followed by a subsequent culturing at 18 ℃ for 18 h as outlined previously. *E. coli* cells were harvested at 4,500 rpm, 4℃ for 20 min. The cell pellets were lysed in lysozyme buffer with sonication and then clarified by ultracentrifugation at 18,000 rpm, 4 ℃ for 30 min. Next, the supernatants were incubated with Ni-NTA agarose resin (YEASEN Biotech, China) for 20 min at 4 ℃, before being processed through an

affinity chromatography column (Beyotime, China) for protein elution. The eluted proteins were dialyzed against $K^+$-free 25 mM Tris/MES buffer (pH 7.4) overnight at 4 ℃ and subsequently stored at 4 ℃ until used.

The Absorption spectra of RGEPOs were measured using a UV-Vis-NIR Spectrophotometer (Shimadzu, Japan), covering a wavelength range from 260 to 700 nm. The fluorescence spectra were collected with FLS1000 Spectrofluorometer controlled by Fluoracle spectrometer operating software (Edinburgh Instruments, UK). The EC were calculated employing the alkaline denaturation method, as previously described [14]. Briefly, the biosensor solution was rapidly mixed with 2M NaOH at a 1:1 volume ratio. The absorption spectrum was then promptly recorded for subsequent comparison with the solution before adding NaOH. The absolute fluorescence QY and fluorescence lifetime measurements were conducted using the FLS1000 spectrometer (Edinburgh Instruments, UK) which was outfitted with an integrating sphere accessory, strictly adhering to the guidelines provided by the manufacturer.

Rapid kinetic measurements detailing the interactions between RGEPOs and potassium ions were conducted utilizing a Stopped Flow Spectrometer (SX 20, Applied Photophysics UK). The fluorescence detection parameters included an excitation wavelength of 560 nm, with a bandwidth of 1 nm, and an emission filter set to 580/10 nm. Fluorescence was recorded for 5,000 time points over 250 s in three sequential repeats, respectively.

To conduct pH titrations, protein solutions were adjusted to various pH levels (from 3 to 10) using Hydrion Buffer Chemvelopes (Hydrion, USA) in the presence and absence of 150 mM KCl. To assess $K^+$ specificity, proteins were introduced into buffers containing distinct ions: LiCl (5 mM), NaCl (15 mM), KCl (150 mM), RbCl (5 mM), CsCl (5 mM), $NH_4Cl$ (5 mM), $MgCl_2$ (20 mM), $CaCl_2$ (20 mM), and $ZnCl_2$ (10 μM) (all from Sigma-Aldrich, USA). For evaluating $K^+$ interactions under nonisotonic conditions, protein solution was diluted into a series of $K^+$ buffers ranging from 0 to 500 mM within Tris/MES buffers (pH 7.4). For evaluating $Ca^{2+}$ interactions under nonisotonic conditions, protein solution was diluted into a series of $Ca^2$ buffers ranging from 0 to 1.07 mM within Tris/MES buffers (pH 7.4). For isotonic $K_d$ measurement, protein solutions were added into isotonic solutions characterized by varied ratios of KCl to N-methyl-D-glucamine (NMDG, Sigma-Aldrich, USA), all prepared in Tris/MES buffers (pH 7.4), with $K^+$ concentrations examined from 0 to 250 mM. NMDG was employed to maintain consistent osmotic pressure throughout the experiments. The fluorescence responses of these protein solutions were measured using a Thermo Scientific Varioskan LUX spectrophotometer, setting the excitation wavelength at 560 nm and measuring emission between 580 and 640 nm.

## Concurrent one- and two-photon spectral characterization

For fluorescence measurements, the samples were diluted to have optical densities less than 0.1. Fluorescence emission and excitation spectra were measured with an LS55 spectrofluorimeter (Perkin Elmer). Fluorescence quantum yields ($\varphi$) were determined using the relative method with cresyl violet in methanol as a standard ($\varphi = 0.54$ [69]). Fluorescence lifetimes were measured on dilute solutions with a Digital Frequency Domain system ChronosDFD (ISS) appended to a PC1 (ISS, Champaigne, IL) spectrofluorometer. The excitation was modulated with multiple harmonics in the range of $10 - 300$ MHz. The modulation ratio and phase delay curves were fitted to model functions corresponding to a double-exponential fluorescence decay with Vinci 3 software (ISS). Absorption spectra were measured with a Lambda900 spectrophotometer (Perkin Elmer). Maximum EC of the anionic and neutral states of chromophore, $\varepsilon_A$ and $\varepsilon_N$, were obtained using gradual alkaline titration method [70]. Here, we define $\varepsilon_A$ and $\varepsilon_N$ as an optical density of 1-molar concentration of either anionic or neutral form of chromophore in 1-cm cuvette. The previously defined in the Characterization of RGEPOs in Solution section effective EC correspond to products $\varepsilon_{eff,A} = \varepsilon_A \rho_A$ and $\varepsilon_{eff,N} = \varepsilon_N \rho_N$, where $\rho_A$ and $\rho_N$ are the fractional concentrations of the chromophore anionic and neutral states, respectively. These concentrations, at neutral pH were obtained from the relative optical densities of the corresponding absorption peaks and the known EC (using Beer's law).

The methods and protocols of the two-photon characterization were described in previous work [71]. Briefly, the two-photon absorption spectra were measured as corrected fluorescence excitation spectra by stepwise tuning the wavelength of a femtosecond laser (DeepSee Insight, MKS—Spectra-Physics) with a LabView program and collecting

fluorescence signal at each wavelength with a PC1 spectrofluorimeter. The power dependence of fluorescence signal was checked for several excitation wavelengths, and only the data points where it was quadratic were selected and presented. For the spectral shape measurements, a combination of short-pass filters (770SP and 680SP) was used to block the scattered laser light. The cross-section $\sigma_{2,A}$ of the anionic form was measured as described in recent work [72] by exciting the two-photon fluorescence at 1,060 nm and one-photon fluorescence at 532 nm. At these wavelengths, only the anionic form gets excited. Rhodamine 6G (Rh6G) in methanol was used as a reference standard. The fluorescence was collected through a combination of 561LP and 770SP filters. To obtain the two-photon excitation spectra in units of molecular brightness, the corrected (for the spectral variations of laser properties) unscaled 2PE spectra were scaled to the product $F_{2,A} = \varphi_A \, \sigma_{2,A} \, \rho_A$ evaluated at 1,060 nm. To measure the signal-to-background ratio upon saturating with $K^+$, the fluorescence signals (either under 1P or 2P excitation) were measured in the $K^+$-free and $K^+$-saturated samples with the same concentration of protein and under the same conditions.

## Imaging of RGEPOs in HEK293FT cells

HEK293FT cells (ATCC ACS-4500, USA) were cultured in Dulbecco's modified Eagle's medium (DMEM, Gibco) supplemented with 10% fetal bovine serum (FBS, CellMAX, China) and 200 U/mL penicillin–streptomycin (YEASEN Biotech, China) at 37 °C with 5% $CO_2$. Cells were seeded on 14 mm coverslips (Biosharp, China) pre-coated with Matrigel (Animal Blood Ware, China) in 24-well plates before transfection. Transfection of HEK293FT cells was performed employing Hieff Trans (YEASEN Biotech, China) according to the manufacturer's protocol. Briefly, 500 ng of plasmids and 1 μl of transfection reagent were added to each well. Imaging was conducted 24–36 h posttransfection, utilizing an inverted wide-field Nikon Eclipse Ti2 microscope, equipped with a SPECTRA III light engine (Lumencor, USA) and an Orca Flash4.0v3 camera (Hamamatsu, USA), and operated via NIS-Elements AR software, utilizing 20× NA0.75 or 10× NA0.45 objective lens.

In the process of intracellular validation, the transfected cells were initially incubated in a basal buffer solution composed of 1.5 mM $CaCl_2$, 1.5 mM $MgSO_4$, 1.25 mM $NaH_2PO_4$, 26 mM $NaHCO_3$, and 10 mM D-Glucose (pH 7.4) with 15 μM valinomycin (Aladdin Biochemical Technology Co., China) and 5 μM CCCP (Shanghai Macklin Biochemical Co., China) for 15 min at 22–25 °C. For gramicidin-based experiments, cells were incubated in the same basal buffer supplemented with 10 μg/mL gramicidin (MCE, USA) for 15 min at 22–25 °C. Following incubation, cells were subjected to perfusion with a series of potassium-enriched buffers, with concentrations varying from 0 to 200 mM, each prepared in the basal buffer via an automated perfusion system, operating at a flow rate of 1 mL/min. The methodology ensured a stable extracellular environment for the cells, negating the need for manual intervention in applying the solutions, as previously described [14]. For the purpose of extracellular validation, cell culture medium was replaced with basal buffer before prior to imaging. For the initial screening experiments shown in Fig 2E and 2F, 200 mM KCl was manually applied to cells via precision pipetting. For the titration experiments (Fig 2H and 2I), a gradient of potassium ion concentrations ranging from 0 to 50 mM was delivered using the automated perfusion system to examine the fluorescence response under varying extracellular potassium conditions. To evaluate photoactivation behavior, transfected HEK293FT cells were continuously imaged under green-light excitation (555/20 nm), with five additional blue-light stimulations (470/28 nm, 2.1 mW/mm², 5 s pulse).

## Culture and imaging in primary neurons and astrocytes

All animal maintenance and experimental procedures for mice were conducted according to the Westlake University animal care guidelines, and all animal studies were approved by the Institutional Animal Care and Use Committee of Westlake University under animal protocol (AP#19-044-KP-2).

Cortical astrocytes and hippocampal neurons were harvested from postnatal day 0 C57BL/6 pups, without distinction of sex, obtained from the Animal Facility at Westlake University. Dissected tissues were digested using 0.25% Trypsin-EDTA (YEASEN Biotech, China) for 10 min at 37 °C, a process that was subsequently halted by the addition of advance medium (Gibco, USA) containing 10% FBS (Gibco, USA).

The dissociated cortical astrocytes were seeded on 60 mm culture dishes (Thermo Fisher Scientific, USA) pre-coated with Matrigel (Animal Blood Ware, China). Half of the culture medium was refreshed every 3–4 days with DMEM (Gibco, USA) supplemented with 10% FBS (CellMAX, China) every 3–4 days. For transfection procedures, astrocytes were further sub-cultured in 24-well plates (NEST, China), also pre-coated with Matrigel (Animal Blood Ware, China), followed by transfection with the pAAV-CAG-RGEPO1/2 plasmids using Hieff Trans (YEASEN Biotech, China) according to the manufacturer's instructions. For localization experiments, astrocytes were co-transfected with pAAV-CAG-RGEPO1 and pDisplay-Igκ-EGFP using the same Hieff Trans reagent and protocol as described above.

The dissociated neurons were plated on 14 mm coverslips (Biosharp, China) pre-coated with Matrigel (Animal Blood Ware, China). Neuronal cultures were maintained in Neurobasal-A medium (Gibco, USA) supplemented with 10% FBS (Gibco, USA) and 2% B27 supplement (Gibco, USA), with medium half changes every 2–3 days. At 3 days in vitro (DIV), AraC (0.002 mM, Sigma-Aldrich, USA) was introduced to selectively inhibit astrocytes proliferation. Cultured neurons were transduced with rAAV2/9 virus (rAAV2/9-hSyn-GCaMP6f, titer, >$10^{12}$ viral genome per ml, Shanghai Sunbio Medical Biotechnology; rAAV2/9-hSyn-RGEPO1, titer, >$10^{12}$ viral genome per ml, BrainVTA (Wuhan) Co.; rAAV2/9-hSyn-RGEPO2, titer, >$10^{12}$ viral genome per ml, Shanghai Sunbio Medical Biotechnology) at 5 DIV. For RGEPOs and GINKO2 expression, cultured neurons were transfected at DIV 5–6 with GINKO2 in combination with either RGEPO1 or RGEPO2 using the calcium phosphate method according to the manufacturer's protocol (Invitrogen, USA). Briefly, 750–1,000 ng of plasmid DNA was added to each well of a 24-well plate. Measurements on neurons were conducted between DIV 10 and 13. The cell culture conditions and imaging setup for astrocytes and neurons followed the protocols outlined in the section describing HEK293FT cells. For dual-color imaging, the green and red fluorescence channels were acquired independently to minimize potential artifacts in the red channel caused by blue light-induced photoactivation of RGEPOs (S11 Fig).

## Virus injections and immunohistochemistry

The neonatal intraventricular injections of rAAV2/9 (rAAV2/9-hSyn-GCaMP6f, titer, >$10^{12}$ viral genome per ml, Shanghai Sunbio Medical Biotechnology; rAAV2/9-hSyn-RGEPO1, titer, >$10^{12}$ viral genome per ml, BrainVTA (Wuhan) Co.; rAAV2/9-hSyn-RGEPO2, titer, >$10^{12}$ viral genome per ml, Shanghai Sunbio Medical Biotechnology) were performed for characterization in brain tissue [73]. Briefly, viruses were injected pan-cortically into pups at postnatal day 0, regardless of sex, with a Hamilton microliter syringe. To induce hypothermic anesthesia, the pups were put on foil with ice underneath until they stopped responding to gentle squeezing on the limbs. 0.5 µL virus solution (titer: >$10^{12}$ v.g. mL$^{-1}$) supplemented with 0.1% FastGreen dye (F7252, Sigma-Aldrich) was injected into each hemisphere. The pups were then moved onto a heating pad maintained at 37 °C for a 5-min recovery. After regaining response to gentle squeezing, the pups were returned to the home cages. To prevent cannibalism, the whole injection process for each pup was strictly controlled within a few minutes.

To characterize the expression of RGEPO1 and RGEPO2 in fixed tissue, the mice at 1–1.5 months old were deeply anesthetized using 1% sodium pentobarbital and transcardically perfused with pre-chilled PBS followed by 4% paraformaldehyde (PFA; Beyotime, China). Brain tissue was harvested and postfixed in 4% PFA for 4–6 h at 4 °C. Followed by rinsing in PBS three times, the coronal brain tissue sections at a thickness of 50 µm were sliced using a VT1200S Vibratome (Leica Microsystems). The free-floating slices were incubated with Nissl (dilution ratio 1:1,000; Invitrogen, USA) diluted in PBS for 20 min at 22–25 °C and then washed three times for 5 min each with PBS. Following mounted with prolonged gold antifade reagent (ThermoFisher; USA), the brain slices were imaged at Nikon Spinning Disk Field Scanning Confocal Systems (CSU-W1 SoRa) with 10× and 40× objective lenses.

## Acute brain slice preparation and imaging

Neonatal AAV injected mice were deeply anesthetized with 1% pentobarbital sodium and then perfused transcardially with cold (2–4 °C) dissecting solution containing (in mM): 213 sucrose, 5 KCl, 1.4 $NaH_2PO_4$, 26 $NaHCO_3$, 1 $CaCl_2$, 3 $MgCl_2$, and 10 D-(+)-glucose saturated with 95% $O_2$ and 5% $CO_2$, pH 7.4 adjusted with NaOH, 300–310 mOsm/L. After the brain

tissue was swiftly trimmed, coronal slices with 350 μm thick were cut off using a vibratome (VT1000S, Leica Microsystems, Germany) and then transferred into a prewarmed chamber (BSC-PC, Harvard Apparatus, USA) for 1-h incubation at 34 °C with oxygenated regular aCSF containing (in mM): 124 NaCl, 5 KCl, 1.4 $NaH_2PO_4$, 26 $NaHCO_3$, 2.4 $CaCl_2$, 1.2 $MgCl_2$, and 10 D-(+)-glucose saturated with 95% $O_2$ and 5% $CO_2$ (pH 7.4 adjusted with NaOH, 300–310 mOsm/L), followed by at least 1 h recovery at room temperature (21–25 °C) before imaging. The imaging setup for acute brain slices followed the protocols described for HEK293FT cells at 22–25 °C. All perfusion buffers were saturated with 95% $O_2$ and 5% $CO_2$.

## Virus injection and cranial window implantation

Mice received cortical injection of a mixture of rAAV2/9-hSyn-GCaMP6f and rAAV2/9-hSyn-RGEPO1/2 virus on day P0. The procedure and conditions were similar to that described for acute brain slice imaging. Seven to eight weeks postinjection, the mice were ready for craniotomy. Using a skull drill, a 3.0-mm diameter craniotomy was created above the center of the primary somatosensory cortex (AP: −0.9 mm, ML: −3.0 mm approximately). The skull was carefully removed with surgical forceps. A coverslip was then implanted onto the craniotomy region and a custom titanium headplate was then secured to the skull with cement. To reinforce the window, a layer of denture base resin was applied over the dental cement and allowed to dry for 10 min. The mice were then returned to their home cages and given at least 4 days to recover postsurgery.

## Two-photon in vivo imaging

For two-photon imaging, awake animals were head-fixed onto a spherical treadmill after recovery. Imaging was performed using a commercial Olympus FVMPE-RS microscope with a 25× water immersion objective lens. To induce seizures, kainic acid (20 mg/kg; cat#487-79-6, MedChemExpress, USA) was diluted in PBS and injected intraperitoneally into the mice. Functional images (512×512 pixels, 0.994 μm/pixel) were acquired at 3 Hz using a resonant scanner to capture fluorescence dynamics. To image GCaMP6f calcium signals, laser was operated at 930 nm through band-pass filter BA495–540. To image RGEPOs, the laser was chosen at 1,080 nm to allow separation and efficient laser transmission through the beam splitter (LCDM690-1050), with emission light filtered through band-pass filter BA575–645.

## Molecular dynamics

We carried out classical MD simulations for both RGEPO1 and RGEPO2 systems at different ion concentrations, as described below.

In order to prepare the systems, we used AlphaFold2 version 1.5.5, where the NMR structure of the potassium binding domain (PDBID: 7PVC) and the crystal structure of mCherry (PDB ID: 2H5Q) were used to validate the final result [14,74,75]. After that, MolProbity version 4.5.2 [76] and PDB2PQR web server version 3.5.2 [77] were used to obtain the proper protonation states for protein residues in each system at pH 7. Then we arranged simulations under different conditions as described below.

To run free diffusion simulations in 20 mM ionic concentration conditions for RGEPO1 and RGEPO2, we randomly added 11 potassium ions to neutralize the systems and achieve a 20 mM concentration. We then solvated the systems using TIP3P water model with a 14 Å water pad from the surface of the protein [78]. To generate the topology and coordinate files, we used tleap from AMBER20 [79]. The force field parameters used were ff14SB for the protein [80] and custom-made parameters for the chromophore using AutoParams [81]. After heating to 300 K, all systems were equilibrated by restraining the protein and chromophore while leaving the solvent unrestrained. We performed a series of iterative relaxations starting at 100 kcal/mol and incrementally decreasing the restraints towards 0, with the density approaching 1.0 g/cm$^{-3}$. A 200 ns classical MD simulation was then run on the equilibrated system using the Langevin thermostat [82] with a 5 ps$^{-1}$ friction parameter in the NVT ensemble with pmemd.cuda in Amber20. The trajectories were run an additional two times for 100 ns each to confirm statistical convergence.

To run in 300 mM ionic concentration system simulations, we carried out MD simulations for both RGEPO1 and RGEPO2 with a 300 mM ionic concentration. For the potassium system, we randomly added 38 potassium ions along with the appropriate number of chloride ions (27) to neutralize the system. Similarly, for the calcium system, we added 38 calcium ions with corresponding chloride ions (65) to maintain electroneutrality. The simulations were conducted following the same procedure as in 20 mM ionic concentration systems.

To run in 300 mM ionic concertation system with bound $K^+$ at Kbp site (PDB ID −7PVC), we carried out MD simulations for both RGEPO1 and RGEPO2 with same procedure. The only difference is that one of the randomly added $K^+$ ions was placed at the initially reported binding site of Kbp (PDB ID −7PVC) to see the stability and the potential effect on the sensitivity of fluorescence. In total, we ran 1,600 ns of MD.

### Data analysis and statistics

Data were analyzed using NIS Elements Advance Research software (versions 5.21.00 and 5.30.00), Excel (Microsoft 2021MSO), OriginPro (2019b, OriginLab), Fiji ImageJ (2.9.01/1.53t.), Clustal Omege (online version) and ESPrint 3.0 [83]. Analysis of all fluorescence traces was conducted as follows: cells and a neighboring cell-free region were manually selected using NIS Elements Advance Research software, and fluorescence measurements were performed for each region of interest (ROI). Fluorescence from an RGEPO-free region was subtracted from the cell fluorescence to account for background, except for the data from acute brain slices. Fluorescence change ($\Delta F/F$) was then calculated using the formula $[(F_{max} - F)/F]$, in which $F$ is the baseline fluorescence intensity. In addition, for acute brain slice data (Fig 6F and 6I), RGEPOs fluorescence traces were corrected for photobleaching by subtracting baseline fluorescence traces that were fit to an exponential function. Except for Fig 3F (RGEPO2), C for different $K^+$ were fitted with Hill equation ($F = F_{max} * [ligand]^n/(EC50^n + [ligand]^n)$), where $F$ is the fluorescence intensity for a defined concentration of ligand, $F_{max}$ is the maximum fluorescence intensity, $EC_{50}$ is the half-maximum effective concentration, and $n$ is the Hill coefficient. For Fig 3F (RGEPO2), dose–response curve was fitted with DoseResp equation ($F = A1 + (A2 - A1)/(1 + 10^{((LOGx0 - x) * p)})$), where $F$ is the fluorescence intensity at a given $K^+$ concentration, $A1$ and $A2$ are the minimum and maximum fluorescence intensities, respectively, LOGx0 corresponds to the log of the $EC_{50}$, $p$ is the Hill slope, and $x$ is the potassium concentration. The data are represented as mean ± SD. Computational data of the systems were analyzed using cpptraj [84] (Root mean square deviation, hydrogen bonding percentages, distance measurements). For the visualization, and to calculate electrostatic charge distribution of the surface of systems, we used ChimeraX [85]. To create the movie of potassium-bound RGEPO systems, we used VMD [86].

For in vivo two-photon imaging of RGEPOs, we used NormCorre [87] for piece-wise rigid motion correction of the images. Neuronal ROI segmentation was performed on the average projection image of GCaMP6f using CellPose [88] and then further refined and corrected using the average projection image of RGEPOs to determine the co-expressing population. Images were first denoised using publicly available Python code of SUPPORT to enhance the seizure-related neuronal activities and spreading wave feature of the recordings [89]. Fluorescence traces were then extracted from the ROIs using ImageJ by averaging pixel intensities for each frame. The fluorescence response of cells was represented as a $\Delta F/F$ trace, where $F$ is the average of the first 100 frames from the recording. Photobleaching of each trace was corrected by fitting using a single-exponential decay. For temporal analysis of wave arrival in cell populations (Fig 7G and 7H), we extracted the wave period $\Delta F/F$ trace of ROIs from wave period time frames and determined the onset time frame of rise periods using first and second derivative thresholding after smoothing using a sliding window. The wave onset time of each cell was normalized across the population by referring zero to the earliest detected onset in the field.

### Supporting information

**S1 Data. Summary of numerical values used for data plots and statistical analysis.**
(XLSX)

**S1 Table.  Properties of genetically encoded fluorescent potassium biosensors.**
(DOCX)

**S2 Table.  Photophysical properties of RGEPO1 sensor.**
(DOCX)

**S3 Table.  Photophysical properties of RGEPO2 sensor.**
(DOCX)

**S1 Fig.  Alignment of amino acid sequences of Ec-Kbp and Hv-Kbp with five homologs obtained from a metage-nomic BLAST search.** Residues within the red boxes with white text indicate the strict identity of amino acids, while those within a blue frame denote similarity among the homologs. Labels for β-sheet-forming regions and α-helix-forming regions are provided as arrows/straight lines and curved lines, respectively. The symbol 'η' denotes a 310 helix, and 'TT' indicates strict β-turns. Residues located within 3.0 Å of the potassium ion in the Ec-Kbp structure (PDB: 7PVC) are marked with asterisks ($N=6$ amino acids). The amino acid numbering follows that of Ec-Kbp. Alignment of protein sequence was conducted via Clustal Omega and subsequently shown with Ec-Kbp structure by ESPript 3.0.
(TIFF)

**S2 Fig.  Fluorescence response of potassium green fluorescence sensors generated by swapping Ec-Kbp with Hv-Kbp and five newly identified homologs in solution.** The homologs were inserted into mNeoGreen following the previous KRaION1's design [4] and titrated with 230 mM $K^+$ in solution. The underlying numerical data for this figure can be found in S1 Data.
(TIFF)

**S3 Fig.  Amino acids sequence of prototype and relative mutants. (A)** Amino acids sequence of RGEPO-prototype sensor with linker sequences Linker1: P and Linker2: SAP. **(B)** The mutants were validated in mammalian cells, corresponding to the red-highlighted dots in Fig 1E.
(TIFF)

**S4 Fig.  Representative single-plane confocal fluorescence images of leader-sequence variants and single-trial optical traces.** Left, single plane fluorescence images of RGEPOs with five different leader sequences expressed in HEK293FT cells in response 20 mM KCl ($n=2$ field of views (FOVs) from 1 culture). Scale bars, 50 μm. Middle, magnified images in $K^+$ free and $K^+$ binding state ($n=3$ cells from 2 FOVs over 1 culture for each). Scale bars, 10 μm. Right, single-trial optical traces of RGEPOs with different leader sequence upon stimulation of 200 mM KCl, which is shown in the Fig 2E ($n=31, 13, 22, 22, 24$ cells from 1 culture for Asp, Azu, CRFR, EtbR, and Igκ, respectively). The underlying numerical data for this figure can be found in S1 Data.
(TIFF)

**S5 Fig.  Validation of membrane localization of Asp-RGEPO0.5 expressed in HEK293T cells.** Fluorescence images and membrane localization analysis of Asp-RGEPO0.5 expressed in HEK293FT cells. Membrane-targeted GFP (Igκ-GFP) was co-expressed to label the plasma membrane. Left, single-plane confocal fluorescence images of HEK293T cells expressing the RGEPO0.5 (red) and EGFP (green) ($n=12$ cells from 2 independent transfections from one culture). Right, normalized linecut (shown as white dashed line on the left) plots of fluorescence signals measured in both the red and green channels. Scale bars, 10 μm. The underlying numerical data for this figure can be found in S1 Data.
(TIFF)

**S6 Fig.  Validation of the top three mutants in the cytoplasm of HEK293FT cells.** Validation of the top three mutants, which exhibited the highest extracellular responses, in the intracellular environment using a perfusion system ($n=34, 44,$

33 cells from 1 culture each, respectively). Dot, individual data point for single cells; bar, mean; error bar, SD. The underlying numerical data for this figure can be found in S1 Data.
(TIFF)

**S7 Fig. Amino acids sequence of prototype, RGEPO0.5, RGEPO1, and RGEPO2.** The sequences derived from mApple and the potassium-binding domain are highlighted in red and blue boxes, respectively. The nonred shaded areas indicate mutations acquired during the engineering process.
(TIFF)

**S8 Fig. Validation of RGEPOs' reversibility and representative images of RGEPO2 expressed in the cytoplasm of HEK cells. (A)** Fluorescence intensity change ($\Delta F/F0$) time course of RGEPO1 with stimulation by a series of K$^+$ buffer on HEK293T cells ($n = 5$ cells from 1 culture), data are expressed as mean (solid line) and SD (shaded area). (B) Representative images of RGEPO2 expressed in the HEK293FT cells ($n = 32$ cells from 1 cultures). Scale bar, 10 μm. Images were obtained via an inverted wide-field Nikon Eclipse Ti2 microscope with 20× NA0.75 objective lens. **(C)** Fluorescence intensity change ($\Delta F/F$) time course of RGEPO2 with stimulation by a series of K$^+$ buffer on HEK293T cells using valinomycin and CCCP ($n = 34$ cells from 1 culture), data are expressed as mean (solid line) and SD (shaded area). The underlying numerical data for this figure can be found in S1 Data.
(TIFF)

**S9 Fig. Time course of fluorescence intensity changes of RGEPOs in HEK293FT. (A, B)** Time courses of fluorescence intensity change ($\Delta F/F$) of RGEPO1 on HEK293FT cells stimulated with a series of K$^+$ buffers. Each trace represents data from an independent culture ($n = 33$ and 25 cells, respectively). Data are shown as mean±SD. **(C–E)** Time courses of fluorescence intensity change ($\Delta F/F$) of RGEPO2 in HEK cells stimulated with a series of K$^+$ buffers in the presence of 10 μg/mL gramicidin. Each trace represents data from an independent culture ($n = 32$ and 36 cells, respectively). Data are shown as mean±SD. **(F)** Representative images of expression and fluorescence change of RGEPO1 in response to 20 mM K$^+$. Scale bar, 10 μm. **(G)** Fluorescence intensity change ($\Delta F/F$) time course of RGEPO1 with stimulation by a series of K$^+$ buffers on HEK293T cells ($n = 18$ cells from 1 culture), data are expressed as mean±SD. The titration was performed by manual addition of each buffer. **(H)** Plot of normalized $\Delta F/F$ against different K$^+$ concentrations fitted using nonlinear fitting (Hill) for the data shown in panel G. The underlying numerical data for this figure can be found in S1 Data.
(TIFF)

**S10 Fig. Time course of fluorescence intensity changes of RGEPO2 in HEK293FT in the presence of valinomycin and CCCP. (A)** Representative images of expression and fluorescence change of RGEPO2 in response to 200 mM K$^+$. Scale bar, 10 μm. **(B–E)** Time courses of fluorescence intensity change ($\Delta F/F$) of RGEPO2 in HEK cells stimulated with a series of K$^+$ buffers in the presence of valinomycin and CCCP. Trace represents data from 3 independent culture ($n = 32, 22, 10$ and 16 cells, respectively). Data are shown as mean±SD. **(F)** Plot of normalized $\Delta F/F$ against different K$^+$ concentrations fitted using nonlinear fitting (Hill) for the data shown in panels C, D,and E ($n = 80$ cells from 3 independent cultures). The underlying numerical data for this figure can be found in S1 Data.
(TIFF)

**S11 Fig. Blue-light photoactivation experiments of RGEPOs in HEK293FT cells.** Photoactivation of RGEPO1 ($n = 10$ cells) and RGEPO2 ($n = 8$ cells) was assessed in HEK293FT cells by continuous imaging with green-light excitation (555/20 nm), combined with five additional blue-light stimulations (470/28 nm, 2.1 mW/ mm$^2$, 5 s pulse, as indicated). The underlying numerical data for this figure can be found in S1 Data.
(TIFF)

**S12 Fig. Steady-state excitation fluorescence spectra of RGEPOs at 0 and 150 mM potassium at pH = 7.4.** The underlying numerical data for this figure can be found in S1 Data.
(TIFF)

**S13 Fig. One-photon (1P) and two-photon (2P) spectral characterization of RGEPOs.** Fluorescence emission and excitation spectra of K⁺-free (blue lines) and K⁺-saturated (red lines) states of RGEPO1: **(A, B)** and RGEPO2: **(E–F)**. Left side of the figure (panels A, C, E, G) corresponds to spectra obtained under one-photon excitation. Excitation spectra are shown as solid lines, and emission spectra as dashed lines: (A) and (E). Panels (C) and (D) represent the ratio of fluorescence intensities of the saturated state versus free state as a function of laser excitation wavelength. Right side of the figure (panels B, D, F, H) corresponds to the two-photon excitation spectra. Panels (D) and (H) represent the ratio of fluorescence intensities of saturated versus free states as a function of two-photon laser excitation wavelength. The underlying numerical data for this figure can be found in S1 Data. The underlying numerical data for this figure can be found in S1 Data.
(TIFF)

**S14 Fig. Ca²⁺ titration of RGEPOs in the range of 0 to 1.07 mM in solution.** Ca²⁺ titration of RGEPOs in the range of 0–1.07 mM in solution ($n = 3$ technical replicates; mean ± SD). The underlying numerical data for this figure can be found in S1 Data.
(TIFF)

**S15 Fig. Change in root mean square deviation (RMSD) of RGEPO1 and RGEPO2 systems with time over first 100 ns.** The first two row graphs show the free diffusion simulations of RGEPO1 (left column) and RGEPO2 (right column) at their respective ionic concentrations. The last row graphs represent simulations of RGEPO1 (left) and RGEPO2 (right) with a 300 mM ionic concentration, initiated with a K⁺ bound at the originally reported Kbp.K binding site (PDB ID: 7PVC). The underlying numerical data for this figure can be found in S1 Data.
(TIFF)

**S16 Fig. Comparison of mutation positions with time.** Structure of RGEPO1 (in red) and RGEPO2 (in salmon pink) with mutated positions highlighted in sticks.
(TIFF)

**S17 Fig. Molecular dynamics simulation of Ca²⁺ binding site. (A)** The predicted 3D structure of RGEPO1 is shown in red, with calcium ions represented in light green. Residues surrounding the binding pockets are shown as blue sticks. This binding region is more solvent-exposed compared to the transient pocket observed for K⁺/Na⁺ in RGEPO1 (located ~5.5 Å away). **(B)** Enlarged from (A), showing the main residues at the Ca²⁺ binding pocket. **(C)** Sensing mechanism of RGEPO1 upon Ca²⁺ binding. **(D)** Distance fluctuation between the K317 side chain nitrogen and the phenolate oxygen of the chromophore over time and Ca²⁺-bound intervals are marked with brackets. **(E)** Density plot generated using the distance data in (D), illustrating bound and unbound Ca²⁺ states. **(F)** Number of water molecules within 3.5 Å (orange) and 5 Å (blue) from the chromophore's O-position throughout the simulation. Periods with bound Ca²⁺ are highlighted in pink. **(G)** The predicted 3D structure of RGEPO2 is shown in pink, with calcium ions represented in light green. Residues surrounding the binding pockets are shown as blue sticks. **(H)** Enlarged from (G), showing the main residues at the Ca²⁺ binding pocket. **(I)** Sensing mechanism of RGEPO2 upon Ca²⁺ binding. **(J)** Distance variation between K317 side chain nitrogen and chromophore phenolate oxygen with time and Ca²⁺ bound time period is highlighted with a bracket. **(K)** Density plot made by using the distance variation data of graph (J), showing the bound and unbound variation. **(L)** Variation in the number of water molecules surrounding the chromophore O-position within 3.5 Å (orange) and 5 Å (blue) throughout the simulation trajectory. Time periods during which a Ca²⁺ ion is bound are highlighted in pink.
(TIFF)

**S18 Fig. Possible transient sites of RGEPO1.** Figures a and b show the electrostatic charge distribution on the surface of RGEPO1 (**A**: front view, **B**: back view after a 180° rotation). In these figures, red represents more negatively charged regions, while blue represents more positively charged regions, including both potential transient sites and main binding sites (labeled with an arrow) as observed in free diffusion simulations with $K^+$ shown in green spheres.
(TIFF)

**S19 Fig. Hydrogen bonding analysis of the chromophore in RGEPO1 at 20 and 300 mM $K^+$.** **(A)** Atom names of the chromophore and **(B)** table of hydrogen bonding analysis.
(TIFF)

**S20 Fig. Variation of distance between K317 side chain nitrogen atom and the $O^-$ atom of the chromophore during $K^+$ bound and unbound states.** Graph **(A)** shows the density plot for the variation in distance between the K317 side chain nitrogen and the $O^-$ of the chromophore, where free diffusion of $K^+$ occurs at the transient binding site of RGEPO1 at 20 mM concentration. Graphs **(B)** to **(D)** display the density plots for the free diffusion simulations of $K^+$ at the main binding sites of RGEPO1 and RGEPO2, as labeled. Graphs **(E)** and **(F)** correspond to $K^+$ bound simulations at the initially reported Kbp-K site. In graph (E), the analysis focuses on the free diffusion of $K^+$ at the main binding site, while in graph (F), it pertains to $K^+$ at the initially reported site. The underlying numerical data for this figure can be found in S1 Data.
(TIFF)

**S21 Fig. Variation of number of water molecules at the $O^-$ end of the chromophore.** These graphs depict the change in the number of water molecules around the $O^-$ position within 3.5 Å (orange) and 5 Å (blue) throughout the trajectory. The time periods during which a $K^+$ ion is bound are highlighted in pink. All simulations are for free diffusion of $K^+$, with binding occurring at the main binding site. However, the simulation for RGEPO1 at 20 mM concentration specifically focuses on the transient binding of K+. The underlying numerical data for this figure can be found in S1 Data.
(TIFF)

**S22 Fig. The electrostatic charge distribution of potassium indicators. (A)** The electrostatic charge distribution of RGEPO1 is shown, with red indicating more electronegative regions and blue representing more electropositive regions. **(B)** The electrostatic charge distribution of GINKO1 (genetically encoded potassium ion biosensor) (PDB ID – 7VCM) is shown, with red indicating more electronegative regions and blue representing more electropositive regions. **(C)** The electrostatic charge distribution of the Kbp.K domain (PDB ID: 7PVC) is shown, with red indicating more electronegative regions and blue representing more electropositive regions. The bound $K^+$ is depicted as a purple sphere. **(D)** The structure of RGEPO1 is shown in red, with the transiently bound $K^+$ in a green sphere and the $K^+$ at the Kbp.K site (PDB ID: 7PVC) in a gray sphere.
(TIFF)

**S23 Fig. Representative images of live mouse hippocampal neurons expressing Asp-RGEPO1 and RGEPO2.** Maximum intensity projection confocal images of Asp-RGEPO1 on the extracellular surface and RGEPO2 in the cytoplasm of live cultured neurons and astrocytes in ACSF buffer ($n = 4$ and 17 neurons from 2 independent cultures; $n = 25$ and 17 astrocytes from 2 independent cultures). Scale bars, 10 µm (scale bar of the second image of Asp-RGEPO1on the astrocyte is 30 µm).
(TIFF)

**S24 Fig. Time course of fluorescence intensity changes of RGEPOs and GCaMP6f in the neuron. (A, B)** Time course of fluorescence intensity changes of Asp-RGEPO1 and GCaMP6f with stimulation of 30 mM KCl on neurons. Each trace represents data from an independent culture ($n = 13$ and 4 neurons, respectively). Data are shown as mean ± SD.

**(C, D)** Time course of fluorescence intensity changes of RGEPO2 and GCaMP6f with stimulation of 30 mM KCl in neurons. Each trace represents data from an independent culture ($n = 38$ and 31 neurons, respectively). Data are shown as mean ± SD. The underlying numerical data for this figure can be found in S1 Data.
(TIFF)

**S25 Fig. Time course of fluorescence intensity changes of RGEPOs and GINKO2, respectively. (A)** Representative dual-color wide-field imaging of RGEPO1 and GINKO2 dynamics in dissociated hippocampal neurons with response to 30 mM KCl. **(B)** Time course of fluorescence intensity changes of RGEPO1 and GINKO2 with stimulation of 30 mM KCl in neurons ($n = 22$ neurons from 3 independent cultures). **(C)** Representative dual-color wide-field imaging of RGEPO2 and GINKO2 dynamics in dissociated hippocampal neurons with response to 30 mM KCl. **(D)** Time course of fluorescence intensity changes of RGEPO2 and GINKO2 with stimulation of 30 mM KCl in neurons ($n = 51$ neurons from 3 independent cultures). The underlying numerical data for this figure can be found in S1 Data.
(TIFF)

**S26 Fig. Validation of membrane localization of Asp-RGEPO1 expressed on the extracellular surface of astrocytes.** Fluorescence images and membrane localization analysis of Asp-RGEPO1 expressed on the extracellular surface of astrocytes. Membrane-targeted EGFP (Igκ-EGFP) was co-expressed to label the plasma membrane. Left, single-plane confocal fluorescence images of astrocytes expressing the RGEPO1 (red) and EGFP (green) ($n = 4$ cells from one independent culture). Right, normalized linecut (shown as white dashed line on the left) plots of fluorescence signals measured in both the red and green channels. Scale bars, 20 μm. The underlying numerical data for this figure can be found in S1 Data.
(TIFF)

**S27 Fig. Time course of fluorescence intensity changes of RGEPOs in the astrocytes. (A–C)** Time courses of fluorescence intensity change ($\Delta F/F$) of RGEPO1 on astrocytes stimulated with a series of K$^+$ buffers. Each trace represents data from an independent culture ($n = 11$, 11 and 13 astrocytes, respectively). Data are shown as mean ± SD. **(D, E)** Time courses of fluorescence intensity change ($\Delta F/F$) of RGEPO2 in astrocytes stimulated with a series of K$^+$ buffers. Each trace represents data from an independent culture ($n = 8$ and 14 astrocytes, respectively). Data are shown as mean ± SD. **(F)** Representative fluorescence images of primary astrocyte expressing RGEPO1 in response to 20 mM KCl ($n = 25$ astrocytes from 2 independent cultures). Scale bars, 100 μm, insert, 20 μm. **(G)** Time course of fluorescence intensity changes of RGEPO1 with stimulation by a series of KCl buffers on astrocytes ($n = 6$ astrocytes from 1 culture), data are expressed as mean ± SD. The titration was performed by manual addition of each buffer. **(H)** Plot of normalized $\Delta F/F$ against different K$^+$ concentrations fitted using nonlinear fitting (Hill) for the data shown in panel G. The underlying numerical data for this figure can be found in S1 Data.
(TIFF)

**S28 Fig. Time course of fluorescence intensity changes of RGEPO2 titrated with a series of KCl buffer using valinomycin and CCCP. (A, B)** Time courses of fluorescence intensity change ($\Delta F/F$) of RGEPO2 in the astrocytes stimulated with a series of K$^+$ buffers. Panel (A) represents data from one independent culture ($n = 12$ cells), and panels B and C are from another independent culture ($n = 6$ and 5 cells, respectively). Data are shown as mean ± SD. (B) Plot of normalized $\Delta F/F$ against different K$^+$ concentrations fitted using nonlinear fitting (Hill) for the data shown in panels A, B, and C. The underlying numerical data for this figure can be found in S1 Data.
(TIFF)

**S29 Fig. Image of the cerebral cortex showing the expression of Asp-RGEPO1.** Upper left, single-panel confocal fluorescence image of the cerebral cortex displaying the expression of Asp-RGEPO1 (red) (scale bar = 1,000 μm). Upper right, higher magnification of single panel confocal fluorescence images highlighting relative expression regions (scale

bar = 500 μm). Lower, further magnified multi-panel confocal fluorescence images showing detailed expression regions (scale bar = 50 μm).
(TIFF)

**S30 Fig.  Image of the cerebral cortex showing the expression of RGEPO2 and GCaMP6f.** Upper left, single-panel confocal fluorescence image of the cerebral cortex displaying the expression of RGEPO1 (red) and GCAMP6f (green) (scale bar = 1,000 μm). Upper right, higher magnification of single panel confocal fluorescence images highlighting relative expression regions (scale bar = 200 μm). Lower, further magnified multi-panel confocal fluorescence images showing detailed expression regions (scale bar = 50 μm). In this representative image, the expression of RGEPO2 appears stronger in the right hemisphere and weaker in the left. Although bilateral AAV injections were performed, the observed asymmetry in expression intensity may be due to slight differences during manual injection, such as pipette placement, angle, or tissue backflow. Additionally, the selected section may have included more of the expression region in the right hemisphere. These technical differences are commonly encountered in intracerebral viral delivery and do not affect our main conclusions.
(TIFF)

**S31 Fig.  Imaging KA-induced seizures using RGEPOs. (A)** Fluorescence response traces from a mouse co-expressing GCaMP6f and RGEPO1, with vertical projections of neuronal fluorescence profiles during KA-induced seizures and subsequent spreading waves, $n = 160$ neurons from one mouse. Right panels represent zoomed-in view of the vertical line profile of GCaMP6f and RGEPO2, highlighting the onset of the spreading wave. **(B)** Temporal analysis of the spreading wave of GCaMP6f and RGEPO1 following seizure activity. Left: pseudo-colored mask of individual neurons overlaid on the average projection image. Right: same mask as left, colored based on fluorescence peak time, $n = 111$ neurons from one mouse. **(C)** Fluorescence response traces from a mouse co-expressing GCaMP6f and RGEPO2, with vertical projections of neuronal fluorescence profiles during KA-induced seizures and subsequent spreading waves, $n = 160$ neurons from one mouse. **(D)** Temporal analysis of the spreading wave of GCaMP6f and RGEPO2 following seizure activity, $n = 29$ neurons from one mouse. **(E, F)** Similar to (C, D), but for another mouse co-expressing GCaMP6f and RGEPO2. $n = 36$ neurons from one mouse. The gray shaded box indicates the period of acute motion observed along the z-axis. Colored arrow indicates the estimated direction of wave propagation. Scale bar: 100 μm. The underlying numerical data for this figure can be found in S1 Data.
(TIF)

**S1 Movie.  Time-lapse imaging of Asp-RGEPO1 on the extracellular surface of HEK293FT cells in response to 20 mM KCl.** Imaging was conducted at 20 ℃–25 ℃ using confocal microscopy.
(AVI)

**S2 Movie.  Time-lapse imaging of RGEPO2 expressed in the cytoplasm of HEK239FT cells in response to 200 mM KCl in the presence of gramicidin.** Imaging was conducted at 20 ℃–25 ℃ using an inverted wide-field Nikon Eclipse Ti2 microscope.
(MP4)

**S3 Movie.  Time-lapse imaging of RGEPO2 and GCaMP6f in the cytoplasm of cultured neurons in response to 500 μM glutamate.** Imaging was conducted at 20 ℃–25 ℃ using an inverted wide-field Nikon Eclipse Ti2 microscope.
(MP4)

**S4 Movie.  Time-lapse imaging of dual-color fluorescence of RGEPO2 and GCaMP6f in the neuronal cytoplasm in response to 30 mM KCl stimulation.** Imaging was conducted at 20 ℃–25 ℃ using an inverted wide-field Nikon Eclipse Ti2 microscope, capturing dynamic changes in both potassium and calcium signals over time.

(MP4)

**S5 Movie. Time-lapse imaging of Asp-RGEPO1 on the extracellular surface of cultured astrocytes in response to a series of KCl buffer ranging from 0 to 50 mM.** Imaging was performed at 20 ℃–25 ℃ using an inverted wide-field Nikon Eclipse Ti2 microscope.
(MP4)

**S6 Movie. Time-lapse imaging of RGEPO2 localized in the cytoplasm of cultured astrocytes in response to a series of KCl buffer ranging from 0 to 200 Mm without valinomycin or CCCP.** Imaging was performed at 20 ℃–25 ℃ using an inverted wide-field Nikon Eclipse Ti2 microscope.
(AVI)

**S7 Movie. Single-plane dual-color imaging of GCaMP6f and RGEPO1 activity from L2/3 neurons of primary somatosensory cortex in awake mouse during KA-induced seizures.** Images were acquired using Olympus FVMPE-RS 2P microscope. Imaging conditions: 512 × 512 pixels, 0.994 µm/pixel, acquired at 3 Hz. Scale bar, 100 µm.
(MP4)

**S8 Movie. Single-plane dual-color imaging of GCaMP6f and RGEPO2 activity from L2/3 neurons of primary somatosensory cortex in awake mouse during KA-induced seizures.** Images were acquired using Olympus FVMPE-RS 2P microscope. Imaging conditions: 512 × 512 pixels, 0.994 µm/pixel, acquired at 3 Hz. Scale bar, 100 µm.
(MP4)

## Acknowledgments

We are grateful to the Flow Cytometry Core Facility at Westlake University and the Laboratory Animal Resources Center. We also thank Zhong Chen from Westlake University for their assistance with spectroscopic measurements; M.H. Liao and G.C. Fang from Imaging Core at Westlake University for their technical support with the Nikon Confocal microscope. We thank the WSU Supercomputing Grid for computational support. We thank Minho Eom Dr. Young-Gyu Yoon lab for helping with in vivo image processing.

## Author contributions

**Conceptualization:** Fedor V. Subach, Kiryl D. Piatkevich.

**Data curation:** Vishaka Pathiranage, Mikhail Drobizhev, Kiryl D. Piatkevich.

**Formal analysis:** Kiryl D. Piatkevich.

**Funding acquisition:** Kiryl D. Piatkevich.

**Investigation:** Lina Yang, Shihao Zhou, Xiaoting Sun, Kiryl D. Piatkevich.

**Methodology:** Lina Yang, Shihao Zhou, Xiaoting Sun, Hanbin Zhang, Cuixin Lai, Chenlei Gu, Mikhail Drobizhev, Kiryl D. Piatkevich.

**Project administration:** Kiryl D. Piatkevich.

**Resources:** Kiryl D. Piatkevich.

**Software:** Vishaka Pathiranage.

**Supervision:** Alice R. Walker, Kiryl D. Piatkevich.

**Visualization:** Lina Yang, Vishaka Pathiranage, Shihao Zhou, Alice R. Walker, Kiryl D. Piatkevich.

**Writing – original draft:** Lina Yang, Vishaka Pathiranage, Shihao Zhou, Alice R Walker, Kiryl D. Piatkevich.

**Writing – review & editing:** Kiryl D. Piatkevich.

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
