## [Editor Report · Decision Letter 0]

19 Dec 2024

Dear Dr Piatkevich,

Thank you for submitting your manuscript entitled "Genetically Encoded Red Fluorescent Indicators for Imaging Intracellular and Extracellular Potassium Ions" for consideration as a Research Article by PLOS Biology.

Your manuscript has now been evaluated by the PLOS Biology editorial staff and I am writing to let you know that we would like to send your submission out for external peer review.

Once your full submission is complete, your paper will undergo a series of checks in preparation for peer review. After your manuscript has passed the checks it will be sent out for review. To provide the metadata for your submission, please Login to Editorial Manager (https://www.editorialmanager.com/pbiology) within two working days, i.e. by Dec 21 2024 11:59PM.

Kind regards,

Richard

Richard Hodge, PhD

rhodge@plos.org

PLOS

---

## [Decision Letter · Decision Letter 1]

3 Mar 2025

Dear Kiryl,

Thank you for your patience while your manuscript "Genetically Encoded Red Fluorescent Indicators for Imaging Intracellular and Extracellular Potassium Ions" was peer-reviewed at PLOS Biology. Please accept my sincere apologies for the delays that you have experienced during the peer review process. Your manuscript has now been evaluated by the PLOS Biology editors, an Academic Editor with relevant expertise, and by two independent reviewers.

In light of the reviews, which you will find at the end of this email, we would like to invite you to revise the work to thoroughly address the reviewers' reports.

As you will see, the reviewers are generally positive about the RGEPO indicators and think they will be useful tools for the field. However, the reviewers ask that additional data and clarifications are included to fully characterize the indicators. In addition, Reviewer #2 raises concerns with the overall strength of the comparative benchmarking to existing potassium indicators. He/she notes that the RGEPO1/2 sensors should be compared side-by-side to previously developed tools in mammalian cells.

Given the extent of revision needed, we cannot make a decision about publication until we have seen the revised manuscript and your response to the reviewers' comments. Your revised manuscript is likely to be sent for further evaluation by all or a subset of the reviewers.

**IMPORTANT - SUBMITTING YOUR REVISION**

*Re-submission Checklist*

*Published Peer Review*

*PLOS Data Policy*

*Blot and Gel Data Policy*

Best regards,

Richard

Richard Hodge, PhD

rhodge@plos.org

REVIEWS:

Reviewer #1: In this manuscript, Yang et al. present the development of two red genetically encoded potassium indicators (RGEPOs), RGEPO1 and RGEPO2, for studying K+ dynamics in biological systems. The authors employed a robust strategy combining directed evolution and mammalian cell optimization to efficiently generate these novel sensors. Comprehensive characterization of RGEPOs was performed, along with mechanistic insights into K+ binding through structural analysis. Subsequently, the simultaneous recording of K+ and Ca2+ signals using RGEPOs and GCaMP6f in diverse settings, including cultured neurons, astrocytes, brain slices, and in vivo, demonstrates the significant potential of these tools. The development of RGEPOs provides a valuable toolkit for investigating the roles of potassium in various physiological and pathological processes.

My critiques are as follows:

1. Photoactivation Concerns: Given that RGEPOs are based on mApple, a known concern is potential blue-light-induced photoactivation, which could interfere with the use of blue-light-activated optogenetic tools. Have the authors test whether RGEPOs exhibit such phenomenon? This information is crucial for researchers considering the combined use of these sensors with optogenetics.

2. Refinement of Characterization Data: Certain characterization results could benefit from further optimization for enhanced precision. In Fig 2h, extending the measurement duration under each K+ concentration would ensure the fluorescence intensity reaches a stable level, providing more robust data. In Fig 2i & 5i, a denser sampling of K+ concentrations around the calculated Kd would improve the accuracy of the dose-response curve. Additionally, the fit of RGEPO2 curve in Fig 3f appears suboptimal, suggesting a need for careful evaluation and potential refinement of the EC50 value.

3. Two-Photon Excitation Spectrum: Since two-photon microscopy was utilized (Fig 7), including the two-photon excitation spectrum of RGEPOs in a supplementary figure would be highly beneficial for researchers selecting appropriate laser wavelengths for their experiments.

4. Standardization of Data Presentation: The use of log scale in Fig 2i & 5i while employing linear scale in Fig 2l & 5l may cause confusion. Standardizing the data presentation across these figures would enhance clarity and readability.

Reviewer #2: In this study, Yang et al. developed two novel genetically encoded potassium (K+) indicators (GEPOs), which are based on a single (red) fluorescent protein (FP) and on a bacterial Kpb K+ binding domain (Hv-Kbp).

GEPOs were engineered to monitor the K+ dynamics in cells and tissues in real-time with the aim of generating tools with a higher dynamic range and stronger signals for multiplex imaging. These newly designed sensors would thus extend our existing tool box of K+ indicators including GEPII and KIRIN, which are based on Förster resonance energy transfer (FRET) as well as GINKO (and several other) that employ single fluorophores of different colors.

Based on this background the authors develop a membrane-targeted redGEPO (RGEPO1) designed to detect K+ changes in the extracellular space, while the intracellular expressed redGEPO (RGEPO2) allows examination of K+ dynamics in the cytosol and nucleus.

The authors characterize RGEPO1 and RGEPO2 in solution and record their individual fluorescence spectra and fluorescence intensities in the absence (0 mM) and presence of high (150-300 mM) K+ concentrations. Further, they provide evidence for the pH stability and ion selectivity of the engineered K+ biosensors and perform molecular simulations to predict the binding pockets for K+ ions as well as the dynamic changes in charge distributions due to K+ on/off. Upon expression of the RGEPOs in hippocampal neurons and astrocyte cultures in vitro the responsiveness of the biosensors to KCl (in a defined concentration range) was tested to establish their functionality in living mammalian cells. Also, K+ dynamics were studied in acute brain slices and in parallel to Ca2+ changes recorded by GCaMP6f. Finally, RGEPOs and GCaMP6f were employed during kainic acid (KA)-induced seizures in mice providing evidence that these new class of K+ biosensors may allow examination of disease-relevant changes in ion dynamics during epileptiform activity of neurons.

The data regarding the generation, optimization and application of RGEPO1 and RGEPO2 is - for the most part - presented in a comprehensible manner and the experimental approaches and conclusions are convincing. Yet, I have some important points that refer to the validation of the sensors in the mammalian cells systems utilized that should be addressed prior to publication:

major

1) The newly developed RGEPO1 and 2 sensors should be compared side-by-side to existing K+ biosensors such as GEPII or GINKO in a relevant mammalian cell setting e.g. primary neurons. This will allow the reviewer to assess whether two main objectives of the study (besides multiplex analyses) were achieved as the authors intended to establish K+ sensors with a higher dynamic range and stronger signals.

2) In general, the number of biological replicates is rather low for several experiments. Also, it is not always clear how many technical / biological replicates were included in the analysis. For example, the number of independent cultures is presently only n=1 or 2 in Fig. 5 and n=1 mouse was assessed for Fig. 7. It is important to increase the number of cultures / mice at least to n = 3. As for Figure 7, it is essential to corroborate the in vivo data with further replicates, otherwise it is not possible for the reviewer to assess whether the findings shown are coincidental or can be reproduced are therefore of biological significance.

3) During optimization of the biosensors, the highest K+ concentration applied varied from 150 mM to 300 mM. Why?

4) Figure 2a and line 139-141: The authors choose to treat cells simultaneously with valinomycin, CCCP and 200 mM K+. Emptying of intracellular K+ by a K+ ionophore (f.ex. Gramicidin) would result in decreased intracellular K+ levels, following treatment with high K+ concentrations should result in steep increase of ΔF/F. This is something that should be tested to estimate the effect of the intracellular K+ ions on the sensors.

5) Line 180 and Supplementary Figure 8a: In the introduction, the extracellular concentration of K+ was mentioned (3.5 to 5 mM). However, during the perfusion experiment, the authors used higher extracellular K+ concentrations (7-9 mM). Why? How does the physiological K+ range of 3-5 mM alter the fluorescence signals? Additionally, the authors state that the fluorescence signals return to baseline levels. The ΔF/F0 intensity curve, however, seems to decline even below baseline levels. How is this explained?

6) Line 181-182 and Figure 2d, g, h: Representative images in Figure 2d and Figure 2g show the membrane-located RGEPO1 at 0 mM and 20 mM K+. In Figure 2h, the maximal K+ concentration plotted corresponds to 50 mM and the change in ΔF/F was calculated to 286 % for 50 mM. It would be more consistent if representative images, curves and calculations are comparable within one figure.

Similarly, in line 384 and Figure 5h, the maximal K+ concentration applied is 20 mM and ΔF/F calculations were performed for this concentration. Supplementary Movie 5, however, presents the addition of 50 mM KCl. Please revise to show the same maximal concentration for the representative images, movies and calculations.

7) Line 203ff: RGEPO1 was characterized in solution without the ASP leader or PDGFβ anchor. In the experimental set-up, especially in the in vitro and in vivo experiments, RGEPO1 was used with the ASP leader and PDGFβ anchor to target the plasma membrane of the cells. The missing biophysical characterizations of the RGEPO1 fusion protein must be provided.

8) Line 230-231 and Figure 3e: The authors described the ion selectivity of RGEPO1 and RGEPO2, which is important and highly appreciated. However, the data shows an influence of Ca2+ on RGEPO fluorescence intensities. As K+ and Ca2+ play an important role in cellular ion homeostasis and influence each other, the performed molecular dynamic simulations should also examine the Ca2+ binding sites in the sensors. Along these lines it is unclear whether and how parallel analysis of RGEPO and GCaMP6f signals affect each other during multiplexing. Hence, some representative experiments (e.g. 5a-d) must be reproduced with RGEPO (without GCaMP6f) and vice versa with GCaMP6f (in the absence of RGEPO).

9) Figure 5c and Figure 5e: In both panels, the expression of RGEPO2 in neurons is presented. However, the expression pattern appears to be different as in Figure 5c, RGEPO2 is expressed in the cytoplasm and nucleus of the neurons (soma and dendrites) whereas in Figure 5e, the expression is mainly restricted to the nucleus (?) and absent in the dendrites. Please clarify!

10) Line 362-365 and Figure 5d: Glutamate stimulation provoked a K+ efflux in neurons. The Glutamate-provoked Ca2+ dynamics should be shown. Also, the glutamate-induced K+ efflux has already been studied by others using the FRET-based K+ sensor GEPII and this should be fairly mentioned in the MS (doi: 10.1096/fj.202002308RR).

11) Line 368-376 and Figure 5e: As Ca2+ may directly affect the fluorescence signals of the RGEPO biosensors, it is necessary to chelate Ca2+ and to represent the K+ dynamics recorded by RGEPO2 under these conditions. Additionally, the authors state that membrane depolarization occurs in response to the K+ stimulus. Has this statement been validated by using e.g. voltage-sensitive dyes?

12) Line 381 and Supplementary Figure 17: Please confirm the membrane localization of ASP-RGEPO1 in astrocytes by a plasma membrane counterstain. This is important as in the respective figure, astrocytes seem to express membrane-located RGEPO1 in the cytoplasm.

13) Line 413 and Supplementary Figure 20, 21: The expression of RGEPOs is mainly recognized in the right hemisphere even if the authors mentioned in the methods part that the injection was performed in both hemispheres. Is there any explanation for this?

minor

1.) The authors have to exclude several spelling mistakes in the manuscript. As an example: line 128: performance; Line 179: RGEPO; Figure 6f: RGEPO2 is indicated in the x-axis even if the experiment was performed with RGEPO1; Line 354, Figure 5a: GcaMP6f is not highlighted in the panel; Line 1189 and Supplementary Figure Legend 8c: RGEPO1 described instead of RGEPO2. Please correct!

2.) The terminology used for the HEK cells is inconsistent throughout the manuscript. The cells are either described by HEK293FT, HEK or HEK293T. As different variants of HEK293 cells are used in research, please specify the variant in the whole manuscript or introduce an abbreviation.

3.) The effects of kainic acid on hippocampal neurons have recently been monitored using the GEPII K+ biosensor (10.1038/s42003-023-05387-9) in vitro. This is something that may be interesting for the discussion of the present study.

---

## [Decision Letter · Decision Letter 2]

22 Jul 2025

Dear Kiryl,

Thank you for your patience while we considered your revised manuscript "Genetically Encoded Red Fluorescent Indicators for Imaging Intracellular and Extracellular Potassium Ions" for publication as a Research Article at PLOS Biology. This revised version of your manuscript has been evaluated by the PLOS Biology editors, the Academic Editor and the original reviewers.

Based on the reviews, I am pleased to say that we are likely to accept this manuscript for publication, provided you satisfactorily address the remaining comment from Reviewer #1 and the following data and other policy-related requests that I have provided below (A-G):

(A) We routinely suggest changes to titles to ensure maximum accessibility for a broad, non-specialist readership. In this case, we would suggest a minor edit to the title, as follows. Please ensure you change both the manuscript file and the online submission system, as they need to match for final acceptance:

“Sensitive red fluorescent indicators for real-time visualization of potassium ion dynamics in vivo”

(B) Please move the following declarative statement from the conflicts of interest statement to the financial disclosure in the online submission form:

(C) You may be aware of the PLOS Data Policy, which requires that all data be made available without restriction: http://journals.plos.org/plosbiology/s/data-availability. For more information, please also see this editorial: http://dx.doi.org/10.1371/journal.pbio.1001797

-Supplementary files (e.g., excel). Please ensure that all data files are uploaded as 'Supporting Information' and are invariably referred to (in the manuscript, figure legends, and the Description field when uploading your files) using the following format verbatim: S1 Data, S2 Data, etc. Multiple panels of a single or even several figures can be included as multiple sheets in one excel file that is saved using exactly the following convention: S1_Data.xlsx (using an underscore).

-Deposition in a publicly available repository. Please also provide the accession code or a reviewer link so that we may view your data before publication.

Figure 1B, 1E, 2B, 2E-F, 2H-I, 2K-L, 3A-H, 5B, 5D, 5F, 5H-I, 5K-L, 6E-F, 6H-I, S2, S4, S5, S6, S8A, S8C, S9A-D, S9F-G, S10B-F, S12. S13A-H, S14, S15, S20A-F, S21, S24A-D, S25B, S25D, S27A-E, S27G-H, S28A-D

(D) Please also ensure that each of the relevant figure legends in your manuscript include information on *WHERE THE UNDERLYING DATA CAN BE FOUND*, and ensure your supplemental data file/s has a legend.

(E) Please ensure that you are using best practice for statistical reporting and data presentation. These are our guidelines https://journals.plos.org/plosbiology/s/best-practices-in-research-reporting#loc-statistical-reporting and a useful resource on data presentation https://journals.plos.org/plosbiology/article?id=10.1371/journal.pbio.1002128

- If you are reporting experiments where n ≤ 5, please plot each individual data point.

(F) Please ensure that your Data Statement in the submission system accurately describes where your data can be found and is in final format, as it will be published as written there.

(G) Per journal policy, if you have generated any custom code during the course of this investigation, please make it available without restrictions. Please ensure that the code is sufficiently well documented and reusable, and that your Data Statement in the Editorial Manager submission system accurately describes where your code can be found.

We expect to receive your revised manuscript within two weeks.

*Published Peer Review History*

*Press*

Best regards,

Richard

Richard Hodge, PhD

rhodge@plos.org

Reviewer remarks:

Reviewer #1 (Yulong Li, identifies himself): In the revised submission, the authors addressed all of my concerns and suggestions. In response to many valuable comments from the Reviewer 2, the authors further improved the manuscript and convincingly demonstrated that RGEPOs are excellent tools for studying K+ dynamics both in vitro and in vivo. I believe the manuscript is now ready for publication.

Minor point: "RGPO2Cin" in the legend of newly added Supplementary Figure 13 should be "RGEPO2 in".

Reviewer #2: The authors have answered to all my concerns and suggestions appropriately. The manuscript as greatly improved.

---

## [Editor Report · Decision Letter 3]

12 Aug 2025

Dear Kiryl,

On behalf of my colleagues and the Academic Editor, Polina Lishko, I am pleased to say that we can accept your manuscript for publication, provided you address any remaining formatting and reporting issues. These will be detailed in an email you should receive within 2-3 business days from our colleagues in the journal operations team; no action is required from you until then. Please note that we will not be able to formally accept your manuscript and schedule it for publication until you have completed any requested changes.

PRESS

Best wishes, 

Richard

Richard Hodge, PhD

rhodge@plos.org

PLOS
